# SC-OMNIGS: SELF-CALIBRATING OMNIDIRECTIONAL GAUSSIAN SPLATTING

**Huajian Huang**[1][*][†]    **Yingshu Chen**[1][*]    **Longwei Li**[2]    **Hui Cheng**[2]    **Tristan Braud**[1]
**Yajie Zhao**[3]    **Sai-Kit Yeung**[1]
[1] The Hong Kong University of Science and Technology
[2] Sun Yat-sen University    [3] ICT, University of Southern California

## ABSTRACT

360-degree cameras streamline data collection for radiance field 3D reconstruction by capturing comprehensive scene data. However, traditional radiance field methods do not address the specific challenges inherent to 360-degree images. We present SC-OmniGS, a novel self-calibrating omnidirectional Gaussian splatting system for fast and accurate omnidirectional radiance field reconstruction using 360-degree images. Rather than converting 360-degree images to cube maps and performing perspective image calibration, we treat 360-degree images as a whole sphere and derive a mathematical framework that enables direct omnidirectional camera pose calibration accompanied by 3D Gaussians optimization. Furthermore, we introduce a differentiable omnidirectional camera model in order to rectify the distortion of real-world data for performance enhancement. Overall, the omnidirectional camera intrinsic model, extrinsic poses, and 3D Gaussians are jointly optimized by minimizing weighted spherical photometric loss. Extensive experiments have demonstrated that our proposed SC-OmniGS is able to recover a high-quality radiance field from noisy camera poses or even no pose prior in challenging scenarios characterized by wide baselines and non-object-centric configurations. The noticeable performance gain in the real-world dataset captured by consumer-grade omnidirectional cameras verifies the effectiveness of our general omnidirectional camera model in reducing the distortion of 360-degree images.

## 1    INTRODUCTION

The radiance field techniques pioneered by NeRF (Mildenhall et al., 2020) have become an essential paradigm to facilitate scene reconstruction and novel view synthesis. NeRF-based approaches (Barron et al., 2021; Zhang et al., 2020; Barron et al., 2022; Fridovich-Keil et al., 2022; Chen et al., 2022; Müller et al., 2022) implicitly representing the structure and appearance of captured objects generally necessitate a dense set of calibrated images for training. However, NeRF requires comprehensive data capture to reconstruct a scene accurately. 360-degree images can greatly facilitate such data capture. Previous works, such as Huang et al. (2022) and Chen et al. (2023b), have demonstrated the feasibility and efficiency of reconstructing omnidirectional radiance fields in large scenes using sparse and wide-baseline 360-degree image inputs.

Although 360-degree images have shown potential in reconstructing omnidirectional radiance fields, the quality of the reconstructed models is highly dependent on the accuracy of camera intrinsic and extrinsic parameters. Existing methods for recovering 3D information from 360-degree images, including structure-from-motion (SfM) systems (Moulon et al., 2013; Huang & Yeung, 2022), rely on an idealized spherical camera model to describe the mathematical relationship between 2D 360-degree images and 3D world projection. The 360-degree images are typically obtained by stitching multiple wide angle images, inheriting the distortion from each lens and resulting in a complex distortion pattern. The adverse impact of such distortion is neglected in the idealized spherical camera model. Consequently, the inaccurate camera projection modeling leads to poor SfM pose estimation, ultimately compromising the quality of 3D radiance field reconstruction when using real-world data. To enhance system performance under camera perturbation and reduce reliance

---

[*]Equal contribution. [†] Corresponding author: huajian@ust.hk

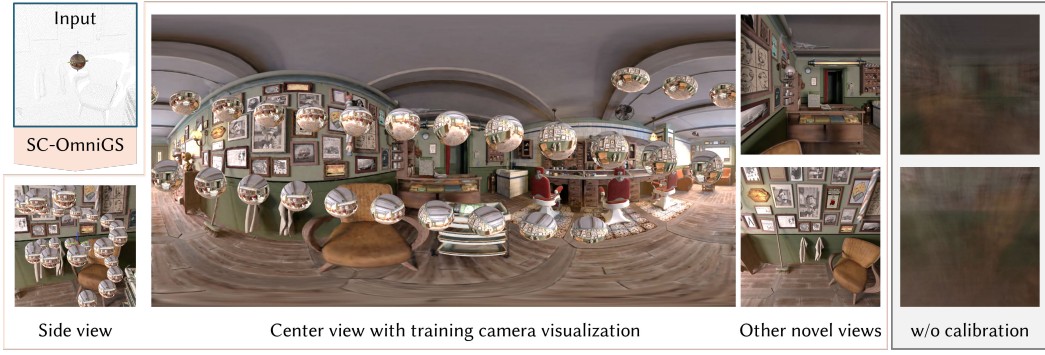

Figure 1: SC-OmniGS jointly optimizes the omnidirectional camera model, poses, and 3D Gaussians using a differentiable omnidirectional rasterizer. It can achieve rapid radiance field reconstruction with no pose prior and render high-fidelity novel views.

on SfM, some approaches (Lin et al., 2021; Jeong et al., 2021; Chen et al., 2023a; Park et al., 2023) have explored radiance field self-calibration, where camera intrinsic and extrinsic parameters are jointly optimized with the radiance field representation. However, these solutions focus on traditional images, using well-established camera models for perspective cameras. A naive approach to self-calibrating the omnidirectional radiance field would consist of projecting the 360-degree images onto cube maps with perspective images. However, this approach undermines the integrity of 360-degree images, leading to increasing optimization complexity and instability (Huang et al., 2024b). Given the lack of camera models accounting for the distortion of 360-degree images and the limitations of existing self-calibration approaches, there is an urgent need for a framework that calibrates the omnidirectional camera model and poses directly.

In this paper, we propose SC-OmniGS, a novel system that self-calibrates the omnidirectional camera model and poses along with omnidirectional radiance field reconstruction. We leverage 3D Gaussian splatting (3D-GS) techniques (Kerbl et al., 2023) to represent radiance fields by a set of 3D Gaussians with explicit positions, covariances, and spherical harmonic coefficients, accelerating the optimization process. In order to realize self-calibrating omnidirectional Gaussian splatting, we adopt a differentiable rasterizer that renders omnidirectional images by splatting 3D Gaussians onto a unit sphere (Li et al., 2024). Crucially, we derive omnidirectional camera pose gradients within the rendering procedure, enabling the optimization of noisy camera poses and even learning from scratch. An example is illustrated in Figure 1. To rectify distortion patterns in the input image, we propose a differentiable omnidirectional camera model comprising a learnable 3D spherical grid to regress the camera distortion. We thus obtain undistorted omnidirectional images by re-sampling input images based on the learned omnidirectional camera model. We jointly optimize 3D Gaussians, camera poses, and camera models by minimizing photometric loss between rendered and undistorted omnidirectional images. The overview of SC-OmniGS framework is demonstrated in Figure 2. Moreover, considering omnidirectional images in the equirectangular projection have an unbalanced spatial resolution, we introduce weighted spherical photometric loss to ensure the spatially equivalent optimization. Furthermore, we apply an anisotropy regularizer to constrain 3D Gaussian scales preventing the generation of filamentous kernels, particularly near the polar areas. To verify the efficacy of SC-OmniGS, we conducted extensive experiments using a synthetic dataset OmniBlender (Choi et al., 2023) and a real-world 360Roam dataset (Huang et al., 2022). The results showed that our proposed system can effectively calibrate the intrinsic model and extrinsic poses of the omnidirectional camera, achieving state-of-the-art performance on the omnidirectional radiance field reconstruction.

To summarize, the main contributions of this work include:

- We proposed the first system for self-calibrating omnidirectional radiance fields, which jointly optimizes 3D Gaussians, omnidirectional camera poses, and camera models.

- We provided the derivation of omnidirectional camera pose gradients within the omnidirectional Gaussian splatting procedure, enabling the optimization of noisy camera poses

and even learning from scratch. It can also facilitate other applications such as GS-based omnidirectional SLAM.

- We introduced a novel differentiable omnidirectional camera model that effectively tackles the complex distortion pattern contained in omnidirectional cameras.

## 2 RELATED WORK

**Omnidirectional Radiance Field**. Neural radiance field (NeRF) (Mildenhall et al., 2020) has emerged as a powerful neural scene representation for novel view synthesis. NeRF represents a scene as a neural network with radiance and opacity outputs for each 3D point. Although most existing radiance field approaches (Chen et al., 2022; Barron et al., 2023; Sun et al., 2022; Xu et al., 2022) can synthesize photorealistic novel views by learning from dense perspective image captures, they tend to suffer from inaccurate geometry reconstruction due to the limited field-of-view coverage and sparse view inputs. To achieve an immersive scene touring with six degrees of freedom (6-DoF), Huang et al. (2022) proposes omnidirectional radiance field learning from sparse 360-degree images with geometry-adaptive blocks, while some previous works incorporate 360-degree 3D priors for better geometry feature learning (Chen et al., 2023b; Kulkarni et al., 2023; Wang et al., 2024). EgoNeRF (Choi et al., 2023) employs quasi-uniform angular grids to enhance performance in egocentric scenes captured within a small circular area. The recent 3D Gaussian splatting (3D-GS) techniques parameterize radiance fields as explicit 3D Gaussians, significantly accelerating rendering and optimization (Kerbl et al., 2023). With the efficient 3D-GS representation, concurrent OmniGS (Li et al., 2024) optimizes 3D Gaussian splats via sparse panorama inputs while 360-GS (Bai et al., 2024) further exploits indoor layout priors for robust structure reconstruction.

While panoramas offer a continuous and wide field of view for omnidirectional optimization, all discussed works focus on radiance field reconstruction merely from known camera parameters, which are vulnerable to inaccurate camera modeling.

**Self-Calibrating Radiance Field**. To simplify the training process of radiance fields and alleviate the reliance on pre-computed camera parameters, some works optimize camera poses or learn poses from scratch during scene reconstruction (Wang et al., 2021; Jeong et al., 2021; Lin et al., 2021). Wang et al. (2021) shows that camera pose and intrinsic parameters can be jointly optimized during NeRF learning for forward-facing scenes. SC-NeRF (Jeong et al., 2021) additionally learns non-linear distortion parameters and introduces a camera self-calibration algorithm for generic cameras during NeRF learning. BARF (Lin et al., 2021) proposes a coarse-to-fine camera registration process from imperfect camera poses for bundle-adjusting NeRFs by gradually activating higher frequency bands of positional encoding. L2G-NeRF (Chen et al., 2023a) introduces an effective local-to-global camera registration strategy with an initially flexible pixel-wise alignment and a frame-wise global alignment. NoPe-NeRF (Bian et al., 2023) employs monocular depth priors for camera estimation with no pose initialization, but it is limited to depth prediction accuracy. For better joint estimation of the scene and camera, CamP (Park et al., 2023) introduces the camera preconditioning technique, which applies a preconditioning matrix to camera parameters before passing them to the NeRF model. Recently, SLAM systems (Huang et al., 2024a; Yan et al., 2024; Matsuki et al., 2024; Keetha et al., 2024) started adopting 3D-GS radiance field for efficient simultaneous localization and photorealistic mapping while the camera intrinsic model is calibrated. Fu et al. (2024) relies on monocular depth estimation for jointly optimizing camera poses and 3D Gaussians.

Existing self-calibrating methods are devised to optimize the radiance field from perspective images. SC-OmniGS is the first work dealing with self-calibration of omnidirectional radiance fields.

**Camera Model.** A camera model is a camera projection function that establishes a mathematical relationship between 2D images and 3D observation. Typically, camera models can be classified into two groups, including parametric camera models, e.g. (Kannala & Brandt, 2006; Usenko et al., 2018) and generic camera models, e.g. (Swaninathan et al., 2003; Schops et al., 2020). Parametric camera models assume in 3D vision that lens distortion is symmetrical radially and use high-order polynomials to approximate models of real lenses. Conversely, generic camera models exploit a mass of parameters to associate each pixel with a 3D ray and calibrate distortion. Recent neural lens modeling (Xian et al., 2023) employs an invertible neural network (Ardizzone et al., 2018-2022) to model lens distortion while its optimization is memory-consuming. In this paper, we propose a generic camera model tailored for the 360-degree camera.

Figure 2: A schematic overview of SC-OmniGS optimization flow.

## 3 PRELIMINARY: 3D GAUSSIAN SPLATTING

3D Gaussian splatting (3D-GS) (Kerbl et al., 2023) represents the scene with a set of 3D Gaussians, of which $i^{th}$ Gaussian is parameterized by 3D position $\mathbf{P}_i$, covariance $\Sigma_i$, opacity $\sigma_i$, and color $c_i$ represented by spherical harmonics (SH) coefficients. The 3D Gaussian reconstruction kernel is formulated as

$$\mathbf{r_{3D}}(\mathbf{P}) = \mathbf{G_{3D}}(\mathbf{P} - \mathbf{P}_i) = exp\{-\tfrac{1}{2}(\mathbf{P} - \mathbf{P}_i)^T \Sigma_i^{-1}(\mathbf{P} - \mathbf{P}_i)\}, \tag{1}$$

where $\mathbf{P} \in \mathbb{R}^3 := (X, Y, Z)^T$ denotes the sampling position in the world space. To render an image, 3D Gaussians are transformed from the world space to the camera space $\{\mathbf{x} := (x, y, z)^T | \mathbf{x} \in \mathbb{R}^3\}$ by a viewing transformation matrix $\mathbf{T} = [\mathbf{R}|\mathbf{t}]$, and $\mathbf{x} = \mathbf{RP} + \mathbf{t}$. 3D Gaussians are then projected onto the image plane $\{\mathbf{u} := (u, v)^T | \mathbf{u} \in \mathbb{R}^2\}$. The projection function $\phi$ for a perspective image is described as

$$\mathbf{u} = \phi(\mathbf{x}) = \begin{bmatrix} f_x x/z + c_x \\ f_y y/z + c_y \end{bmatrix}, \tag{2}$$

where $f_x, f_y$ are focal lengths and $c_x, c_y$ are the principle points of the pinhole camera model. Since this projection process is not affine, the 3D Gaussian reconstruction kernel $\mathbf{r_{3D}}(\mathbf{P})$ cannot be directly mapped to 2D. To address this problem, Zwicker et al. (2002) introduced the local affine approximation of the projection function:

$$\mathbf{u} = \mathbf{u}_i + \mathbf{J}_i \cdot (\mathbf{x} - \mathbf{x}_i) = \phi(\mathbf{RP}_i + \mathbf{t}) + \mathbf{J}_i \cdot (\mathbf{x} - \mathbf{RP}_i - \mathbf{t}). \tag{3}$$

The Jacobian $\mathbf{J}_i$ is defined by the partial derivatives of projection function $\phi$ at point $\mathbf{x}_i$:

$$\mathbf{J}_i = \frac{\partial \phi}{\partial \mathbf{x}}(\mathbf{x}_i) = \begin{bmatrix} \frac{\partial u_i}{\partial x} & \frac{\partial u_i}{\partial y} & \frac{\partial u_i}{\partial z} \\ \frac{\partial v_i}{\partial x} & \frac{\partial v_i}{\partial y} & \frac{\partial v_i}{\partial z} \end{bmatrix}, \tag{4}$$

$$\frac{\partial u_i}{\partial x} = \frac{f_x}{z_i}, \quad \frac{\partial u_i}{\partial y} = 0, \quad \frac{\partial u_i}{\partial z} = -\frac{f_x x_i}{z_i^2}, \quad \frac{\partial v_i}{\partial x} = 0, \quad \frac{\partial v_i}{\partial y} = \frac{f_y}{z_i}, \quad \frac{\partial v_i}{\partial z} = -\frac{f_y y_i}{z_i^2}. \tag{5}$$

According to Eq. 1 and 3, the 2D Gaussian reconstruction kernel is thus calculated by

$$\mathbf{r}_{2D}(\mathbf{u}) = \mathbf{G}_{2D}(\mathbf{u} - \mathbf{u}_i) = \exp\{-\tfrac{1}{2}(\mathbf{u} - \mathbf{u}_i)^T (\mathbf{J}_i \mathbf{R} \Sigma_i \mathbf{R}^T \mathbf{J}_i^T)^{-1}(\mathbf{u} - \mathbf{u}_i)\}. \tag{6}$$

The final rendering color $\mathbf{C}(\mathbf{u})$ of a pixel $\mathbf{u}$ in the image can be computed by volumetric rendering:

$$\mathbf{C}(\mathbf{u}) = \sum_{i \in \mathcal{N}} c_i \alpha_i \prod_{j=1}^{i-1}(1 - \alpha_j), \quad \alpha_j = \sigma_j \cdot \mathbf{r}_{2D}(\mathbf{u}), \tag{7}$$

$$c_i = \sum_{m=0}^{\mathcal{M}} \mathbf{SH}_i^m(dir_i), \quad dir_i = normalize([\mathbf{P}_i - (-\mathbf{R}^T \mathbf{t})]), \tag{8}$$

where $\mathcal{N}$ denotes the set of ordered 3D Gaussians affecting the pixel $\mathbf{u}$ after splatting onto 2D image, while $\mathcal{M}$ is the degree of SH coefficients. $\mathbf{SH}_i^m(\cdot)$ denotes spherical harmonics functions of the normalized viewing orientation $dir_i$.

## 4 METHODOLOGY: SC-OMNIGS

SC-OmniGS is a self-calibrating framework for omnidirectional radiance field reconstruction. It takes multiple 360-degree images without pose information or with noisy pose estimations as input to recover a fine-grained omnidirectional radiance field. We adopt 3D-GS (Kerbl et al., 2023) as the

radiance field representation to achieve fast reconstruction and real-time novel view rendering. Similar to 3D-GS, we initialize the 3D Gaussians from coarse points input obtained from SfM estimation or an omnidirectional depth map. We then jointly optimize 3D Gaussians, the omnidirectional camera model, and poses. The overview of our framework is demonstrated in Figure 2.

In this section, we first revisit omnidirectional Gaussian splatting and introduce a differentiable rasterizer that can render omnidirectional images in the equirectangular projection. In addition, we conduct a mathematical analysis of omnidirectional camera pose derivatives within the rasterizer. Furthermore, we propose a novel omnidirectional camera model to rectify input training images. Finally, the joint optimization is performed to minimize weighted spherical photometric loss and anisotropy loss.

## 4.1 OMNIDIRECTIONAL GAUSSIAN SPLATTING

To develop a universal rasterizer, we adopt an idealized spherical camera model to describe the projection relationship of an omnidirectional camera (Li et al., 2024). Rather than splatting 3D Gaussians onto an image plane, we project them onto a unit sphere and subsequently expand the unit sphere to a 2D image in the equirectangular projection. The projection function for an omnidirectional image, denoted as $\phi^o$, is defined as:

$$\mathbf{u} = \phi^{\mathbf{o}}(\mathbf{x}) = \begin{bmatrix} f_x^o \cdot arctan2(x,z) + c_x^o \\ f_y^o \cdot arcsin(y/d) + c_y^o \end{bmatrix} = \begin{bmatrix} \frac{W}{2\pi} \cdot arctan2(x,z) + \frac{W}{2} \\ \frac{H}{\pi} \cdot arcsin(y/d) + \frac{H}{2} \end{bmatrix}, \tag{9}$$

where $arctan2$ is the 2-argument arctangent function and $d = \sqrt{x^2 + y^2 + z^2}$. $H$ and $W$ denote image height and width respectively. According to Eq. 4, the partial derivatives of projection function $\phi^o$ at point $\mathbf{x}_i$ is $\mathbf{J}_i^o$, and

$$\mathbf{J}_i^o = \frac{\partial \phi^o}{\partial \mathbf{x}}(\mathbf{x}_i) = \begin{bmatrix} f_x^o \cdot \frac{z_i}{x_i^2 + z_i^2} & 0 & -f_x^o \cdot \frac{x_i}{x_i^2 + z_i^2} \\ f_y^o \cdot \frac{x_i y_i}{d_i^2 \sqrt{x_i^2 + z_i^2}} & f_y^o \cdot \frac{\sqrt{x_i^2 + z_i^2}}{d_i^2} & -f_y^o \cdot \frac{z_i y_i}{d_i^2 \sqrt{x_i^2 + z_i^2}} \end{bmatrix}. \tag{10}$$

We substitute $\mathbf{J}_i$ in Eq. 6 and obtain the 2D Gaussian reconstruction kernel for omnidirectional Gaussian splitting:

$$\mathbf{r}_{2D}^o(\mathbf{u}) = \mathbf{G}_{2D}^o(\mathbf{u} - \mathbf{u}_i) = \exp\{-\tfrac{1}{2}(\mathbf{u} - \mathbf{u}_i)^T (\mathbf{J}_i^o \mathbf{R} \Sigma_i \mathbf{R}^T \mathbf{J}_i^{oT})^{-1}(\mathbf{u} - \mathbf{u}_i)\}. \tag{11}$$

Eventually, the rendering color $\mathbf{C^o}(\mathbf{u})$ of a pixel $\mathbf{u}$ in the omnidirectional image can be computed by:

$$\mathbf{C}^o(\mathbf{u}) = \sum_{i \in \mathcal{N}} c_i \alpha_i^o \prod_{j=1}^{i-1} (1 - \alpha_j^o), \quad \alpha_j^o = \sigma_j \cdot \mathbf{r}_{2D}^o(\mathbf{u}). \tag{12}$$

## 4.2 GRADIENTS OF OMNIDIRECTIONAL CAMERA POSE

In addition to backpropagating gradients with respect to 3D Gaussians, our differentiable omnidirectional rasterizer also propagates gradients with respect to world-to-camera transformation metrics $\mathbf{T} = [\mathbf{R}|\mathbf{t}]$ for camera pose optimization. To ensure numerical stability and avoid singularities during optimization, we represent and optimize the transformation matrix in a compact and singularity-free form, which is a 7-dimensional vector comprising a rotation quaternion and translation: $\mathbf{T}' = [\mathbf{q}|\mathbf{t}] = [q_w \, q_x \, q_y \, q_z \, t_x \, t_y \, t_z]$. By applying the chain rule to the rendering function in Eq. 12, the gradients of camera pose can be decomposed into two primary branches: $\frac{\partial \mathcal{L}}{\partial c} \cdot \frac{\partial c}{\partial \mathbf{T}'}$ and $\frac{\partial \mathcal{L}}{\partial \alpha_j^o} \cdot \frac{\partial \alpha_j^o}{\partial \mathbf{r}_{2D}^o} \cdot \frac{\partial \mathbf{r}_{2D}^o}{\partial \mathbf{T}'}$. Since $\frac{\partial \mathcal{L}}{\partial c}$ and $\frac{\partial \mathcal{L}}{\partial \alpha_j^o} \cdot \frac{\partial \alpha_j^o}{\partial \mathbf{r}_{2D}^o}$ have been previously derivated for 3D Gaussian optimization (Kerbl et al., 2023; Li et al., 2024), we further elaborate unsolved parts subsequently.

**Part 1**: $\frac{\partial c}{\partial \mathbf{T}'}$, the gradient of color w.r.t. pose $[\mathbf{q}|\mathbf{t}]$. The view-dependent color of a 3D Gaussian is obtained from spherical harmonics coefficients as depicted in Eq 8. It is related to its normalized viewing orientation. Hence, $\frac{\partial c}{\partial \mathbf{T}'}$ is equal to

$$\frac{\partial c}{\partial \mathbf{T}'} = \frac{\partial c}{\partial dir} \cdot \frac{\partial dir}{\partial \mathbf{T}'} = \frac{\partial c}{\partial dir} \cdot \left[ \frac{\partial dir}{\partial \mathbf{R}} \cdot \frac{\partial \mathbf{R}}{\partial \mathbf{q}}, \quad \frac{\partial dir}{\partial \mathbf{t}} \right]. \tag{13}$$

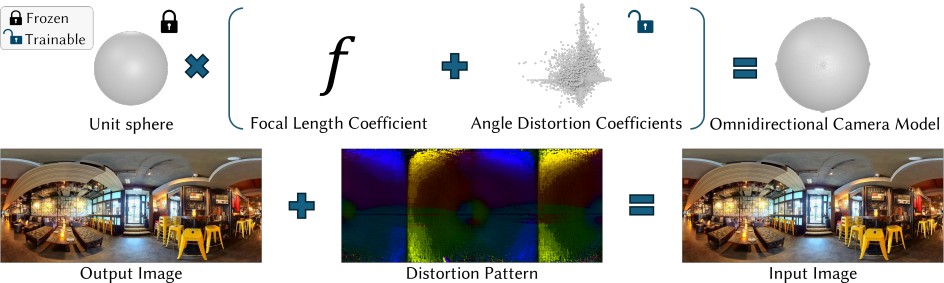

Figure 3: Differentiable omnidirectional camera model.

**Part 2**: $\frac{\partial \mathbf{r}_{2D}^o}{\partial \mathbf{T}'}$, the gradient of 2D Gaussian w.r.t. pose $[\mathbf{q}|\mathbf{t}]$. Camera pose gets involved in the splatting of Gaussian onto 2D omnidirectional images. According to Eq. 9-11,

$$
\begin{aligned}
\frac{\partial \mathbf{r}_{2D}^o}{\partial \mathbf{T}'} &= \left[ \frac{\partial \mathbf{r}_{2D}^o}{\partial \mathbf{u}_i} \cdot \frac{\partial \mathbf{u}_i}{\partial \mathbf{T}'}, \quad \frac{\partial \mathbf{r}_{2D}^o}{\partial \mathbf{J}_i^o} \cdot \frac{\partial \mathbf{J}_i^o}{\partial \mathbf{T}'}, \quad \frac{\partial \mathbf{r}_{2D}^o}{\partial \mathbf{R}} \cdot \frac{\partial \mathbf{R}}{\partial \mathbf{T}'} \right] \\
&= \left[ \frac{\partial \mathbf{r}_{2D}^o}{\partial \mathbf{u}_i} \cdot \frac{\partial \mathbf{u}_i}{\partial \mathbf{x}_i}, \quad \frac{\partial \mathbf{r}_{2D}^o}{\partial \mathbf{J}_i^o} \cdot \frac{\partial \mathbf{J}_i^o}{\partial \mathbf{x}_i} \right] \cdot \left[ \frac{\partial \mathbf{x}_i}{\partial \mathbf{R}} \cdot \frac{\partial \mathbf{R}}{\partial \mathbf{q}}, \quad \frac{\partial \mathbf{x}_i}{\partial \mathbf{t}} \right] + \left[ \frac{\partial \mathbf{r}_{2D}^o}{\partial \mathbf{R}} \cdot \frac{\partial \mathbf{R}}{\partial \mathbf{q}} \right].
\end{aligned}
\tag{14}
$$

### 4.3 OMNIDIRECTIONAL CAMERA MODEL

Omnidirectional cameras, which typically consist of at least two fisheye lenses, capture 360-degree images through image stitching. However, factory calibration prioritizes seamless stitching over rectifying distortion. As such, stitched omnidirectional images inherently retain distortion from the original camera lenses and deviate from ideal spherical camera models. Unfortunately, there is a lack of well-established camera models capable of accurately representing omnidirectional camera distortion, which inevitably compromises 3D reconstruction quality. To address this limitation, we propose the first generic omnidirectional camera model that learns complex distorting patterns through differentiable optimization. Our omnidirectional camera model comprises a frozen unit sphere and trainable focal length coefficient $f_t$ and angle distortion coefficients, as illustrated in Figure 3. For model initialization, we create a spherical grid $\mathcal{S} \in \mathbb{R}^{H \times W \times 3}$ and set the corresponding angle distortion coefficients $\mathcal{D}$ with the same dimension to zeros. The camera ray distortion is then estimated by the Hadamard product of the spherical grid and learned angle distortion coefficients. This approach is more stable than directly learning camera ray distortion. Consequently, the omnidirectional camera model $\Theta$ is defined as:

$$
\Theta := \mathcal{S} \cdot f_t + \mathcal{S} \odot \mathcal{D}.
\tag{15}
$$

Our differentiable camera model is decoupled from the rasterization pipeline, ensuring that it does not compromise the efficiency of the rendering process. By leveraging the learned camera model parameters $\Theta$, we can extract a distortion-free omnidirectional image $I^o$ from the input image using bicubic grid sampling. Please refer to Algorithm 1 for details. The extracted images $I^o$ are then utilized to compute the total loss against the rendered images $I^r$.

### 4.4 JOINT OPTIMIZATION

The optimization in terms of 3D Gaussian, camera pose $\mathbf{T}'$, and camera model $\Theta$ is performed by minimizing the photometric loss, comprising the mean absolute error (MAE) and structural similarity index measure (SSIM) loss. However, the equirectangular image projection is not conformal, as the region deformation increases along parallels towards poles. In other words, similar 3D spatial information would occupy more pixels when projected to the top and bottom areas of the 2D image. To ensure spatially equivalent optimization, we introduce a weighted spherical photometric loss, which is defined as:

$$
\mathcal{L}_{wsp}(I^r, I^o) = \frac{1}{|\mathcal{I}|} \sum_{\mathbf{u} \in \mathcal{I}} \left\{ (1 - \lambda) \left| \hat{I}^r - \hat{I}^o \right|_1 + \lambda(1 - \text{SSIM}(\hat{I}^r, \hat{I}^o)) \right\},
\tag{16}
$$

$$
\hat{I} = wI, \quad w(\mathbf{u}) = \cos\left(v - c_y^o + 0.5\right)/f_y^o
\tag{17}
$$

where $\lambda$ is a hyperparameter, $\mathcal{I}$ represents a set of image pixels, and $w(\cdot)$ is the spherical weights (Sun et al., 2017) used to ensure a spherically uniform sample. In addition, we leverage an anisotropy regularizer to constrain the ratio between the major and minor axis lengths of 3D Gaussians, thereby preventing them from degenerating into filamentous kernels. The anisotropy regularizer is formulated as:

$$\mathcal{L}_{aniso} = \frac{1}{|\mathcal{N}|} \sum_{i \in \mathcal{N}} \left\{ \max(\frac{\max(\mathbf{s}_i)}{\min(\mathbf{s}_i)}, \gamma) - \gamma \right\}, \tag{18}$$

where $\mathbf{s}_i$ is the scaling of 3D Gaussians (Kerbl et al., 2023) and $\gamma$ is the ratio threshold. Overall, the joint optimization objective is:

$$\mathcal{L} = \mathcal{L}_{wsp} + \mathcal{L}_{aniso}. \tag{19}$$

## 5 EXPERIMENTS

### 5.1 EXPERIMENT SETUP

**Implementation Detail.** Our SC-OmniGS implementation is built on Pytorch and CUDA. We utilize Adam optimizer to update trainable parameters. The hyperparameters for 3D Gaussians optimization are set according to the default settings of 3D-GS (Kerbl et al., 2023), with $\lambda = 0.2$ and a total of 30,000 optimization iterations. We set the ratio threshold $\gamma$ to 10. The omnidirectional camera model is shared across all views on individual scene. Moreover, we set the learning rate of the camera model $\Theta$ to 1e-4 and activate the angle distortion coefficients $\mathcal{D}$ using the Tanh function. For simplicity, we fix $f_t$ to 1. The initial learning rates for each camera quaternion $\mathbf{q}$ and translation $\mathbf{t}$ are set to 0.01, with exponential decay to 1.6e-4 and 6e-3, respectively, in 100 steps per camera. When calibrating from scratch, we increase the initial learning rate of $\mathbf{t}$ to 0.1.

**Baselines.** For comparison, we select BARF (Lin et al., 2021), L2G-NeRF (Chen et al., 2023a) and CamP (Park et al., 2023) as SOTA radiance field calibration baselines trained with training cameras initialized with preset perturbations or from scratch with no pose prior. For reference, we also run 3D-GS (Kerbl et al., 2023) and OmniGS (Li et al., 2024) as non-calibration SOTA baselines. However, apart from OmniGS, other baselines devised for perspective images are not compatible with omnidirectional images as input. To accommodate baselines for fair comparisons, we adopted two practices: 1) We converted each omnidirectional image into a cube map consisting of six perspective images, and then we took the cube maps as input to run the open-source systems with default configurations. 2) Following 360Roam (Huang et al., 2022), we replaced the ray sampling functions of NeRF-based methods (BARF, L2G-NeRF, CamP) with omnidirectional ray sampling to support omnidirectional image training and rendering. Additionally, since point cloud initialization is demanded by 3D-GS based methods, we conducted experiments using different initialization strategies to further verify our system's robustness and flexibility.

**Datasets.** We evaluated SG-OmniGS against several SOTA models on datasets of 360-degree images, including eight real-world multi-room scenes from 360Roam dataset (Huang et al., 2022) each with on average 110 training views and 37 test views, and three synthetic single-room scenes from OmniBlender dataset (Choi et al., 2023) each with 25 training views and 25 test views. 360Roam dataset utilizes camera poses estimated by SfM as ground truth and also provides SfM sparse point cloud. OmniBlender dataset provides noise-free camera poses and depth maps.

All methods were run on a desktop computer with an RTX 3090 GPU. We use metrics PSNR, SSIM, and LPIPS for evaluating novel view synthesis. Please refer to Appendix C for details on camera perturbations and experimental setup.

### 5.2 EVALUATION ON SINGLE-ROOM SYNTHETIC DATASET

We conducted experiments on three synthetic scenes from OmniBlender (Choi et al., 2023), namely **Barbershop**, **Classroom**, and **Flat**. As depicted in Table 1, we configured four settings of radiance field calibration,

- Camera poses with perturbation and 3D Gaussians initialized from a single rendering depth map.

- No camera poses prior but 3D Gaussians initialized from a single rendering depth map.

Table 1: Quantitative comparisons on synthetic dataset OmniBlender. Checked "Perturb" indicates perturbed training camera poses for training, † indicates training from scratch. 3D-GS based methods are marked with different point cloud initializations: random sampling (random), projection from an estimated mono-depth (est. depth), or from a rendered mono-depth (render depth). Methods marked with superscript ° are modified via omnidirectional sampling. We mark the best two results in each experiment group with **first** and second .

| On OmniBlender | Perturb | train | | | | | | | | | test | | | | | | | | |
|---|---|---|---|---|---|---|---|---|---|---|---|---|---|---|---|---|---|---|---|
| | | Barbershop | | | Classroom | | | Flat | | | Barbershop | | | Classroom | | | Flat | | |
| | | PSNR↑ | SSIM↑ | LPIPS↓ | PSNR↑ | SSIM↑ | LPIPS↓ | PSNR↑ | SSIM↑ | LPIPS↓ | PSNR↑ | SSIM↑ | LPIPS↓ | PSNR↑ | SSIM↑ | LPIPS↓ | PSNR↑ | SSIM↑ | LPIPS↓ |
| 3D-GS (render depth) | × | 31.308 | 0.922 | 0.093 | 26.489 | 0.782 | 0.248 | 30.274 | 0.882 | 0.149 | 30.526 | 0.912 | 0.101 | 25.794 | 0.766 | 0.262 | 28.357 | 0.869 | 0.161 |
| OmniGS (render depth) | × | 37.270 | 0.971 | 0.040 | 32.565 | 0.857 | 0.161 | 34.484 | 0.928 | 0.081 | 35.485 | 0.965 | 0.043 | 31.552 | 0.846 | 0.164 | 33.477 | 0.922 | 0.083 |
| OmniGS (render depth) | ✓ | 24.155 | 0.830 | 0.268 | 20.175 | 0.699 | 0.399 | 22.904 | 0.813 | 0.285 | 17.717 | 0.595 | 0.446 | 16.917 | 0.561 | 0.484 | 18.768 | 0.700 | 0.372 |
| BARF | ✓ | 28.796 | 0.851 | 0.242 | 25.854 | 0.741 | 0.309 | 28.072 | 0.823 | 0.252 | 23.477 | 0.752 | 0.260 | 25.705 | 0.739 | 0.309 | 22.235 | 0.759 | 0.294 |
| BARF° | ✓ | 30.066 | 0.869 | 0.191 | 29.204 | 0.768 | 0.261 | 31.003 | 0.868 | 0.143 | 29.739 | 0.866 | 0.191 | 28.865 | 0.765 | 0.263 | 30.417 | 0.649 | 0.144 |
| L2G-NeRF | ✓ | 29.023 | 0.858 | 0.222 | 25.585 | 0.729 | 0.325 | 27.970 | 0.825 | 0.243 | 28.749 | 0.856 | 0.224 | 18.064 | 0.597 | 0.408 | 18.937 | 0.713 | 0.353 |
| L2G-NeRF° | ✓ | 30.083 | 0.870 | 0.188 | 29.140 | 0.765 | 0.267 | 31.020 | 0.866 | 0.145 | 29.705 | 0.867 | 0.189 | 28.823 | 0.762 | 0.268 | 30.576 | 0.863 | 0.146 |
| CamP | ✓ | 29.916 | 0.888 | 0.185 | 26.774 | 0.813 | 0.181 | 29.440 | 0.864 | 0.179 | 17.770 | 0.605 | 0.449 | 16.258 | 0.553 | 0.558 | 18.383 | 0.699 | 0.380 |
| CamP° | ✓ | 30.865 | 0.905 | 0.162 | 30.749 | **0.884** | **0.108** | 29.930 | 0.883 | 0.162 | 17.892 | 0.688 | 0.391 | 15.948 | 0.544 | 0.549 | 17.892 | 0.688 | 0.391 |
| Ours (random) | ✓ | 36.255 | 0.960 | 0.066 | 32.764 | 0.848 | 0.185 | 34.476 | 0.918 | 0.094 | 34.719 | 0.957 | 0.062 | 30.659 | 0.827 | 0.189 | 33.344 | 0.912 | 0.096 |
| Ours (est. depth) | ✓ | 36.578 | 0.964 | 0.051 | 33.066 | 0.859 | 0.149 | 33.256 | 0.922 | **0.023** | 34.404 | 0.952 | 0.055 | 30.122 | 0.816 | 0.156 | 31.472 | 0.901 | 0.084 |
| Ours (render depth) | ✓ | **37.612** | **0.978** | **0.028** | **33.075** | 0.875 | 0.127 | **35.240** | **0.941** | 0.063 | **35.612** | **0.972** | **0.030** | **31.151** | **0.853** | **0.132** | **34.129** | **0.935** | **0.065** |
| OmniGS (render depth) | † | 18.507 | 0.689 | 0.542 | 17.160 | 0.622 | 0.555 | 18.758 | 0.747 | 0.395 | 18.431 | 0.678 | 0.542 | 17.120 | 0.611 | 0.556 | 18.728 | 0.744 | 0.395 |
| BARF | † | 27.871 | 0.823 | 0.296 | 24.752 | 0.700 | 0.360 | 27.621 | 0.814 | 0.269 | 18.299 | 0.631 | 0.410 | 16.794 | 0.564 | 0.455 | 20.645 | 0.735 | 0.329 |
| BARF° | † | 27.598 | 0.807 | 0.303 | 25.869 | 0.706 | 0.360 | 28.410 | 0.820 | 0.231 | 27.508 | 0.805 | 0.303 | 25.710 | 0.703 | 0.360 | 28.140 | 0.818 | 0.231 |
| L2G-NeRF | † | 28.300 | 0.840 | 0.255 | 25.623 | 0.731 | 0.324 | 27.911 | 0.820 | 0.258 | 20.165 | 0.679 | 0.317 | 19.461 | 0.621 | 0.377 | 18.921 | 0.714 | 0.359 |
| L2G-NeRF° | † | 28.488 | 0.834 | 0.256 | 26.802 | 0.719 | 0.341 | 29.152 | 0.832 | 0.209 | 28.198 | 0.830 | 0.256 | 26.300 | 0.714 | 0.342 | 28.717 | 0.828 | 0.211 |
| CamP | † | 27.316 | 0.834 | 0.273 | 25.738 | 0.767 | 0.255 | 30.202 | 0.868 | 0.163 | 17.753 | 0.605 | 0.389 | 15.420 | 0.526 | 0.493 | 18.342 | 0.711 | 0.306 |
| CamP° | † | 27.818 | 0.839 | 0.241 | 26.710 | 0.790 | 0.211 | 32.169 | 0.891 | 0.116 | 16.807 | 0.585 | 0.413 | 14.664 | 0.501 | 0.490 | 27.982 | 0.856 | 0.124 |
| Ours (random) | † | 35.196 | **0.953** | **0.075** | 31.082 | 0.833 | 0.203 | 32.614 | 0.903 | 0.111 | 33.422 | **0.944** | 0.084 | 28.971 | 0.806 | 0.214 | 31.673 | 0.895 | 0.114 |
| Ours (est. depth) | † | 35.343 | 0.952 | 0.082 | 32.294 | 0.851 | 0.166 | 32.924 | 0.915 | 0.088 | 33.401 | 0.940 | 0.087 | 29.385 | 0.801 | 0.195 | 31.278 | 0.897 | 0.094 |
| Ours (render depth) | † | **35.601** | 0.961 | 0.060 | **30.815** | 0.846 | 0.173 | 33.064 | 0.910 | 0.110 | **34.368** | 0.956 | **0.063** | **30.212** | **0.837** | **0.176** | **32.424** | **0.906** | **0.112** |

- No camera poses prior but 3D Gaussians initialized from a single estimated depth map.

- No camera poses prior and random 3D Gaussians initialization.

In the first setting, we perturbed the training camera poses using the same preset noises, indicated by "✓" under the "Perturb" column in Table 1. OmniGS is the SOTA method in non-calibration omnidirectional radiance field reconstruction. When the input camera poses contain noticeable perturbance, OmniGS suffers significant performance degradation and struggles to synthesize clear novel views. BARF and L2G-NeRF exhibit acceptable performance with perturbed training cameras. After modifying ray sampling functions, we can effectively improve NeRF-based methods' performance, proving the necessity of properly treating omnidirectional images as a whole. However, we cannot apply a similar modification to 3D-GS based methods. It is non-trivial to achieve omnidirectional radiance field bundle adjustment, while our SC-OmniGS achieves dominant performance, on par with OmniGS trained with ground-truth cameras.

Additionally, we initialized all training cameras at the origin, enabling training the models from scratch without pose priors. This is denoted by a "†" under the "Perturb" column in Table 1. In comparison to all baselines, our SC-OmniGS demonstrates stable and excellent performance. To verify SC-OmniGS flexibility and robustness, we utilized an omnidirectional monocular depth estimation method, e.g. EGformer (Yun et al., 2023), to estimate a depth map of the first image for 3D Gaussians initialization without the necessity of a known camera pose. Despite a slight decrease in rendering quality, the results demonstrate that our method still exhibits significant performance improvements compared to baseline methods. Finally, rather than using the rendered or estimated geometry as the starting point, we randomly sampled 300k points with random colors and positions as the initial 3D Gaussians to run our method. Our method is able to effectively optimize the scene representation, displaying a low sensitivity to initial values.

Figures 4a and 4b display visual comparisons among calibration methods trained from scratch. Based on the conventional pinhole camera model, inaccurate camera optimization for individual perspective views leads to disconnected faces of a cube map, such as red insets of BARF and L2G-NeRF. In contrast, our omnidirectional camera model assists in optimizing cameras with concern about the holistic field of view, achieving a continuous synthesis.

## 5.3 EVALUATION ON MULTI-ROOM REAL-WORLD DATASET

In real-world scenarios, we studied three situations of SC-OmniGS and reported the average metric scores across scenes in Table 2:

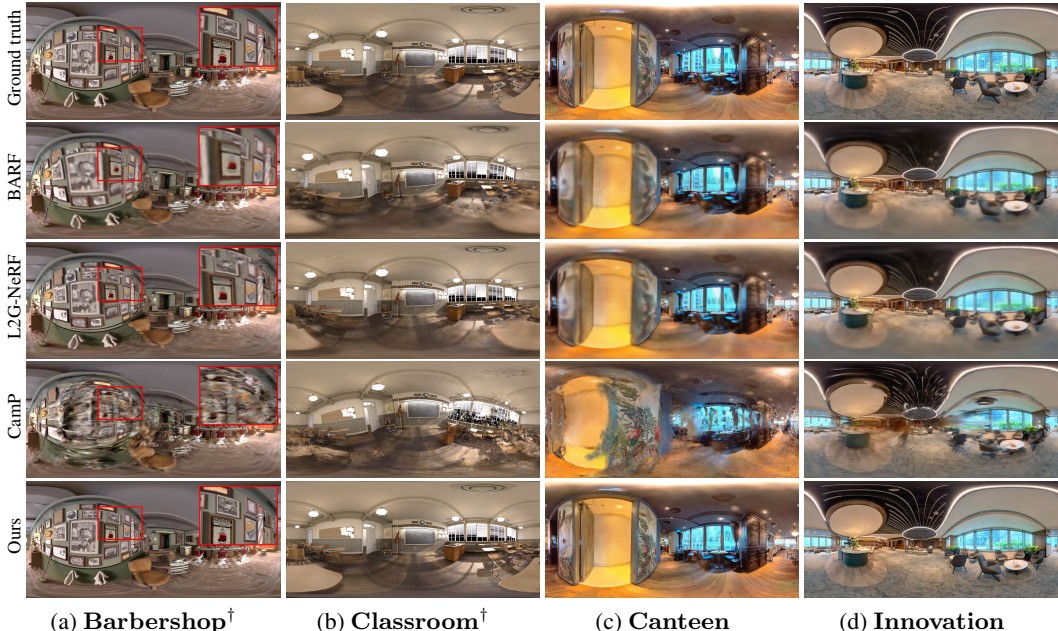

Figure 4: Qualitative comparisons of 360-degree novel views among calibration methods. Our results outperform in both rendering quality and camera accuracy. † indicates training from scratch.

Table 2: Quantitative comparisons on real-world dataset 360Roam. "Point Init" indicates the way of point cloud initialization for 3D-GS based methods, checked "Perturb" indicates perturbed camera poses as inputs, "train" and "test" indicate training and test views, respectively. Methods marked with superscript ° are modified via omnidirectional sampling. We mark the best two results in each experiment group with **first** and second .

| On 360Roam | Perturb | Point Init | train | | | test | | |
|---|---|---|---|---|---|---|---|---|
| | | | PSNR↑ | SSIM↑ | LPIPS↓ | PSNR↑ | SSIM↑ | LPIPS↓ |
| 3D-GS (Kerbl et al., 2023) | × | SfM | 23.943 | 0.744 | 0.223 | 20.791 | 0.684 | 0.261 |
| OmniGS (Li et al., 2024) | × | SfM | 28.517 | 0.861 | **0.137** | 24.212 | 0.768 | **0.176** |
| SC-OmniGS (Ours) | × | SfM | **29.495** | **0.877** | 0.141 | **25.297** | **0.803** | 0.180 |
| OmniGS (Li et al., 2024) | ✓ | SfM | 22.111 | 0.705 | 0.334 | 15.619 | 0.455 | 0.489 |
| BARF (Lin et al., 2021) | ✓ | N/A | 21.699 | 0.594 | 0.465 | 20.200 | 0.572 | 0.481 |
| BARF° (Lin et al., 2021) | ✓ | N/A | 22.136 | 0.575 | 0.492 | 20.484 | 0.546 | 0.510 |
| L2G-NeRF (Chen et al., 2023a) | ✓ | N/A | 21.797 | 0.598 | 0.460 | 20.507 | 0.576 | 0.473 |
| L2G-NeRF° (Chen et al., 2023a) | ✓ | N/A | 22.581 | 0.590 | 0.462 | 20.023 | 0.542 | 0.495 |
| CamP (Park et al., 2023) | ✓ | N/A | 24.592 | 0.735 | 0.264 | 14.253 | 0.438 | 0.573 |
| CamP° (Park et al., 2023) | ✓ | N/A | 26.134 | 0.786 | 0.239 | 13.659 | 0.437 | 0.622 |
| SC-OmniGS (Ours) | ✓ | Random | 28.562 | 0.852 | 0.175 | 24.343 | 0.770 | 0.224 |
| SC-OmniGS (Ours) | ✓ | SfM | **29.232** | **0.872** | **0.147** | **24.910** | **0.790** | **0.188** |

- SfM camera poses without perturbation and 3D Gaussians initialized from SfM point clouds.

- SfM camera poses with perturbation and 3D Gaussians initialized from SfM point clouds.

- SfM camera poses with perturbation and random 3D Gaussians initialization.

Real-world omnidirectional images captured by 360-degree cameras inherit the distortion from each lens and result in a complex distortion pattern. However, most methods leverage an ideal spherical camera model to describe omnidirectional projection while overlooking the impact of 360-degree camera distortion. With our proposed calibration approach, SC-OmniGS can further optimize camera parameters in particular the camera intrinsic model, eventually outperforming the non-calibration method OmniGS trained with SfM cameras, as demonstrated in the first block of Table 2. Under the situation of camera perturbation, SC-OmniGS demonstrates consistent performance across both training and test views, no matter how 3D Gaussians are initialized.

As visualized in Figure 4, our SC-OmniGS also dominates qualitative performance in omnidirectional scenarios. BARF and L2G-NeRF tend to synthesize low-quality and blurry images, while

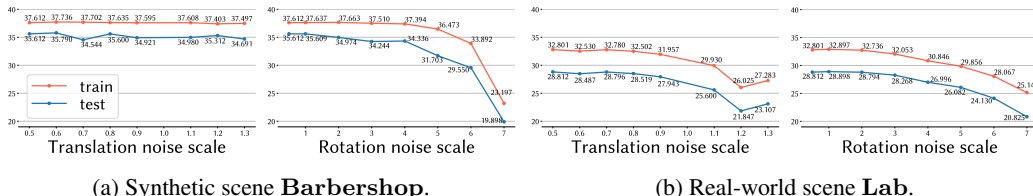

(a) Synthetic scene **Barbershop**.     (b) Real-world scene **Lab**.

Figure 5: Performance with different camera perturbations (PSNR↑). Zoom in for details.

Table 3: Ablation study on scene **Center** of 360Roam, in terms of the optimization of camera pose, camera model, or both. "Perturb" indicates perturbed camera poses, "train" and "test" indicate training and test views, respectively. We mark the best two results with first and second .

| Calibration | w/o Perturb | | | | | | w/ Perturb | | | | | |
| | train | | | test | | | train | | | test | | |
| | PSNR ↑ | SSIM ↑ | LPIPS ↓ | PSNR ↑ | SSIM ↑ | LPIPS ↓ | PSNR ↑ | SSIM ↑ | LPIPS ↓ | PSNR ↑ | SSIM ↑ | LPIPS ↓ |
|---|---|---|---|---|---|---|---|---|---|---|---|---|
| none | 28.728 | 0.848 | 0.170 | 24.264 | 0.763 | 0.213 | 22.740 | 0.717 | 0.372 | 15.597 | 0.510 | 0.553 |
| +camera model | **30.230** | **0.877** | **0.153** | 25.123 | 0.795 | **0.195** | 22.743 | 0.730 | 0.408 | 15.702 | 0.543 | 0.568 |
| +pose | 28.334 | 0.837 | 0.191 | 24.906 | 0.781 | 0.224 | 28.130 | 0.834 | 0.198 | 24.739 | 0.777 | 0.233 |
| +camera model+pose | 30.035 | 0.872 | 0.169 | **25.802** | **0.813** | 0.203 | **29.706** | **0.867** | **0.177** | **25.304** | **0.799** | **0.220** |

CamP generates floating fuzzy artifacts, albeit with some high-frequency details. Please refer to Appendix D for more quantitative and qualitative comparison results.

## 5.4 ROBUSTNESS AND ANALYSIS OF SC-OMNIGS

**Robustness.** To further assess the robustness of our method against varying levels of camera perturbation, we conducted experiments using the same learning rate with increasing scales of translation and rotation noise applied to the training cameras. In Figure 5, we visualize the performance trend depicting the impact of increasing noise scales on the synthetic scene **Barbershop** and the real-world scene **Lab**. In the left charts of Figures 5a and 5b, we fixed the default rotation noise scale and varied translation noise scales, while the right charts represent variable rotation noise scale and fixed translation noise scale. Our camera calibration demonstrates greater robustness to translation errors with only minor degradation compared to rotation errors. Furthermore, when compared to other calibration baselines (see **Barbershop** in Table 1), SC-OmniGS consistently outperforms them with most increased rotation noise scales.

**Ablation study.** As a novel self-calibrating omnidirectional radiance fields method, SC-OmniGS proposed two main components, i.e. a generic omnidirectional camera model and camera pose optimization. To validate the effectiveness of our camera calibration, we conducted ablation studies on a real scene **Center**, with and without perturbation to training cameras. The results are presented in Table 3. When the input camera poses are estimated by SfM without perturbation, we can slightly increase the quality of radiance field reconstruction by camera pose refinement, although its performance gain is not higher than adding an omnidirectional camera model. When trained with pose perturbation, our full model, incorporating both camera model and pose optimization, consistently achieves improvement in both training and test view synthesis.

## 6 CONCLUSION

This paper introduces SC-OmniGS, the first self-calibrating omnidirectional Gaussian splatting system that enables swift and accurate reconstruction of omnidirectional radiance fields. With the differentiable omnidirectional camera model and Gaussian splatting procedure, our approach jointly optimizes 3D Gaussians, omnidirectional camera poses and camera model, leading to robust camera optimization and enhanced reconstruction quality. Extensive experiments validate the effectiveness of SC-OmniGS in recovering high-quality omnidirectional radiance fields, either with noisy poses or without pose prior. Our work offers an efficient and precise omnidirectional radiance field reconstruction for potential applications in virtual reality, robotics, and autonomous navigation.

ACKNOWLEDGMENTS

This research project is partially supported by the Innovation and Technology Support Programme of the Innovation and Technology Fund (Ref: ITS/319/22FP).

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

APPENDIX

## A    SOCIETAL IMPACTS

This research explored the efficient and robust self-calibrating omnidirectional radiance field for large omnidirectional scenarios, experimenting with real-world data captured with the consumer-grade 360-degree camera and synthetic data. It has broad potential impacts and applications in the real world. For example, it supports real-time photorealistic rendering for virtual environments, which enhances virtual immersiveness and enables mixed-reality production. In addition, it can be incorporated into SLAM techniques to upgrade localization robustness.

## B    LIMITATION.

When confronted with challenging omnidirectional scenes, i.e., multi-room-level scenes with sparse discrete views, training from scratch is a challenging task without the assistance of a typical SfM pipeline. We conducted an additional training from scratch experiment using the 360Roam dataset. All self-calibration methods fail to learn radiance fields without any pose priors while our SC-OmniGS is no exception. To address these issues, integrating SC-OmniGS into an omnidirectional SLAM framework is a promising direction, which can be a future work.

## C    EXPERIMENT DETAILS

### C.1    PSEUDO-CODE OF DIFFERENTIABLE OMNIDIRECTIONAL CAMERA MODEL

Algorithm 1 illustrates the backpropagation process and the usage of the proposed generic camera model.

---

**Algorithm 1:** Differentiable Omnidirectional Camera Model

---

**Input:** input image $I$
**/* Initialization */**
$H, W, C \leftarrow$ image dimension of $I$;
$\mathbf{u} \leftarrow$ image pixel coordinates;
$\mathbf{S} \leftarrow \phi'(\mathbf{u})$; // project UV back to camera space
$f_t \leftarrow 1$; // focal length coefficient
$\mathcal{D} \leftarrow$ initialize as zeros in in dimension $(H, W, 3)$;
$\mathcal{D} \leftarrow$ enable gradients; // learnable angle distortion coefficients

**/* Image Undistortion */**
$\mathcal{D} \leftarrow Tanh(\mathcal{D})$; // apply activation function
$\hat{\mathbf{S}} \leftarrow \mathbf{S} \cdot f_t + \mathbf{S} \odot \mathcal{D}$; // Eq. 15
$\hat{\mathbf{u}} \leftarrow \phi(\hat{\mathbf{S}})$; // undistorted UV coordinates
Output undistorted image $I^o \leftarrow grid\_sample(I, \hat{\mathbf{u}})$; // bicubic grid sample

$\mathcal{D} \leftarrow$ backpropagate and update via total loss $\mathcal{L}$;

---

### C.2    DATASETS

**360Roam.**    360Roam (Huang et al., 2022) provides 360-degree captured images by a consumer-grade 360-degree camera for indoor scenes with multiple rooms, and corresponding initial sparse point clouds from SfM. We selected eight scenes with relatively large scales for evaluation, including **Bar, Base, Cafe, Canteen Center, Innovation, Lab,** and **Library**. All data are under CC BY-NC-SA 4.0 license.

**OmniBlender.**    OmniBlender (Choi et al., 2023) contains multi-view 360-degree images rendered from Blender synthetic single indoor scenes under MIT License. It provides ground-truth camera

parameters, and we additionally rendered a ground-truth depth map of each scene to initialize a sparse point cloud for 3D-GS based methods.

The synthetic Blender scene **Classroom** is under CC0 license, **Barbershop** and **Flat** are under CC-BY 4.0 license. All original models can be downloaded in `https://www.blender.org/download/demo-files/`.

### C.3 PERTURBATION DETAILS

In comparison experiments in Sec. 5.3 and 5.2, we add translation noise to SfM or ground-truth camera translation, and multiply rotation by rotation noise. Specifically, we set translation perturbation noise $T_{noise} = \alpha T_{scale} \times inv\_r$, where $\alpha$ is random samples from a uniform distribution over $[-1, 1)$, default $T_{scale} = 0.5$, and $inv\_r$ is the inverse of maximum radius of camera positions for scale normalization. We set rotation perturbation noise $R_{noise} = \beta R_{scale}$, where $\beta$ is normalized rotation direction with dimensional values randomly sampled from a normal distribution over the angle range $[-1°, 1°)$, and default $R_{scale} = 0.5$. Finally, we get preset perturbed translation $\hat{T}$ and rotation $\hat{R}$:

$$\hat{T} = T + T_{noise}, \hat{R} = R \times R_{noise}.$$

In Sec. 5.4 for robustness measurement, we fixed rotation noise scale $R_{scale} = 0.5$ and changed translation noise scale with $T_{scale} \in [0.5, 0.6, 0.7, 0.8, 0.9, 1.1, 1.2, 1.3]$, and also fixed translation with noise scale $T_{scale} = 0.5$ and changed rotation noise scale with $R_{scale} \in [0.5, 1, 2, 3, 4, 5, 6, 7]$.

### C.4 BASELINES

We trained experimental models by all baselines, i.e., 3D-GS (Kerbl et al., 2023), OmniGS (Li et al., 2024), BARF (Lin et al., 2021), L2G-NeRF (Chen et al., 2023a), CamP (Park et al., 2023), using their official published source codes and default training configurations. The baseline authors hold all the ownership rights on their software.

By default, we convert each 360-degree image in Appendix C.2 into a cube map with six non-overlapped $480 \times 480$ perspective images and re-computed six camera parameters. BARF, L2G-NeRF and CamP trained scenes using converted perspective training and test images. In particular, we increase training iterations of 3D-GS to six fold, i.e., 180,000 iterations for each scene for a fair comparison.

In addition, we modified the calibration baselines, i.e., BARF, L2G-NeRF and CamP, by replacing original perspective ray sampling with omnidirectional ray sampling for training and rendering. These modified baselines, OmniGS and our SC-OmniGS trained scenes using resolution $760 \times 1520$ for 360Roam dataset and $1000 \times 2000$ for OmniBlender dataset.

### C.5 RUNTIME

Table 4 reports the quantitative comparisons of training time and inference speed among different methods. On average, for a scene with a GeForce RTX 3090 GPU, BARF trains for over 2 days, L2G-NeRF and CamP for half a day, 3D-GS (six-fold iterations), OmniGS and our SC-OmniGS within 30 minutes. It is noted that SC-OmniGS does not increase much training time with camera self-calibration compared to OmniGS without camera calibration, meanwhile SC-OmniGS supports real-time rendering.

## D MORE EXPERIMENT RESULTS

### D.1 ADDITIONAL ABLATION STUDY

Considering the characteristic of the omnidirectional image, we introduce a weighted spherical photometric loss $\mathcal{L}_{wsp}$ as defined in Eq. 16 for spatially equivalent optimization. Furthermore, we observe that noisy camera poses can lead to the generation of numerous incorrect 3D Gaussians at the beginning of optimization, making it challenging to filter them out during optimization. To address this, we re-initialize the 3D Gaussian with the input coarse points twice, at the $2000^{th}$ and

| Method | Training time | Rendering speed for one panorama (FPS) |
|--------|---------------|----------------------------------------|
| BARF | > 2 days | < 0.05 |
| L2G-NeRF | > 12 hours | < 0.05 |
| CamP | > 12 hours | < 0.2 |
| 3D-GS | 30 mins | > 60 |
| OmniGS | 30 mins | > 60 |
| SC-OmniGS | 30 mins | > 60 |

Table 4: Runtime comparison for methods running on one GeForce RTX 3090 GPU.

Table 5: Ablation study. "Re-init" indicates re-initialization of 3D Gaussians; w/o $\mathcal{L}_{wsp}$ means we disable the spherical weight and calculate classical photometric loss for optimization; "Perturb" indicates perturbation; † indicates training from scratch without pose priors. We mark the best two results with first and second .

| **Classroom** | Perturb | PSNR ↑ | SSIM ↑ | LPIPS ↓ |
|---------------|---------|--------|--------|---------|
| w/o Re-init | † | 29.183 | 0.823 | 0.193 |
| w/o $\mathcal{L}_{wsp}$ | † | 28.225 | 0.811 | 0.192 |
| Ours | † | **30.212** | **0.837** | **0.176** |

$4000^{th}$ iterations. To further verify the effect of the weighted spherical photometric loss and calibration strategy, we conducted additional experiments on **Classroom** as an ablation study. The test view results are reported in Table 5 and Figure 6.

## D.2 MORE QUANTITATIVE AND QUALITATIVE COMPARISONS

We report the complete image quantitative evaluation results on 360Roam dataset in Table 6, and the additional camera pose optimization comparisons in individual scenes in Table 7. Under different scenes and different point cloud initializations, SC-OmniGS outperforms other calibration baselines achieving robust camera calibration capability.

Figures 7-8 supplement some qualitative rendering and depth comparisons among adapted calibration baselines with omnidirectional sampling in the scenes same as Figure 4 in the main manuscript. We should intuitively notice that baselines with omnidirectional sampling render continuous 360-degree views, while our SC-OmniGS still gains the best rendering fidelity and most accurately calibrated cameras. Furthermore, Figures 9-12 visualize more comparison results of novel 360-degree and perspective views among calibration baselines.

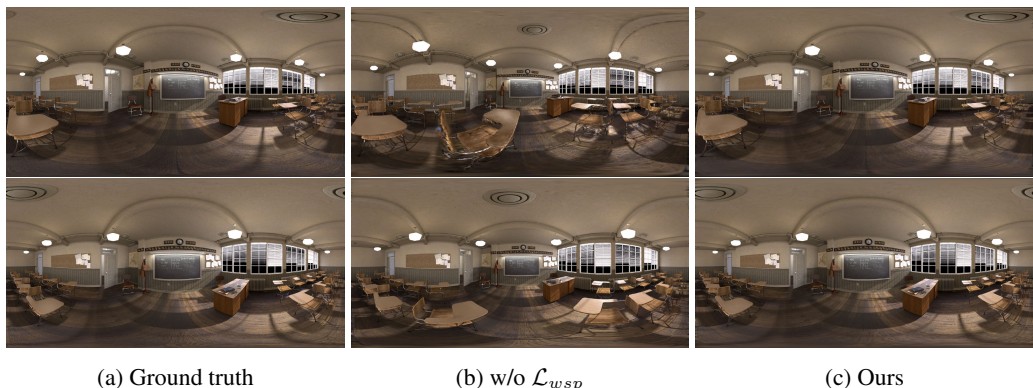

(a) Ground truth        (b) w/o $\mathcal{L}_{wsp}$        (c) Ours

Figure 6: Ablation study of weighted spherical photometric loss $\mathcal{L}_{wsp}$. Without using $\mathcal{L}_{wsp}$, the estimated poses of some cameras suffer obvious errors leading to performance degradation in novel view synthesis.

Table 6: The complete image quantitative evaluation results on real-world dataset 360Roam. Checked "Perturb" indicates perturbed camera poses as inputs, "Point Init" indicates the way of point cloud initialization for 3D-GS based methods, "train" and "test" indicate training and test views, respectively. Methods marked with superscript ° are modified via omnidirectional sampling.

| On 360Roam | | 3D-GS | | OmniGS | | SC-OmniGS | | BARF | | L2G-NeRF | | CamP | |
|---|---|---|---|---|---|---|---|---|---|---|---|---|---|
| | | train | test | train | test | train | test | train | test | train | test | train | test |
| | Perturb | × | | × | | × | | ✓ | | ✓ | | ✓ | |
| | Point Init | SfM | | SfM | | SfM | | N/A | | N/A | | N/A | |
| Bar | PSNR↑ | 20.983 | 18.764 | 24.511 | 21.567 | 25.653 | 22.556 | 19.047 | 18.020 | 19.089 | 18.333 | 22.181 | 13.534 |
| | SSIM↑ | 0.734 | 0.673 | 0.849 | 0.760 | 0.862 | 0.783 | 0.543 | 0.523 | 0.547 | 0.533 | 0.736 | 0.388 |
| | LPIPS↓ | 0.235 | 0.268 | 0.155 | 0.191 | 0.158 | 0.200 | 0.528 | 0.538 | 0.518 | 0.527 | 0.283 | 0.556 |
| Base | PSNR↑ | 23.677 | 20.764 | 28.914 | 24.254 | 30.070 | 25.504 | 20.409 | 19.638 | 20.582 | 19.991 | 23.874 | 13.402 |
| | SSIM↑ | 0.733 | 0.681 | 0.876 | 0.768 | 0.897 | 0.816 | 0.506 | 0.499 | 0.511 | 0.505 | 0.674 | 0.372 |
| | LPIPS↓ | 0.206 | 0.233 | 0.101 | 0.135 | 0.098 | 0.133 | 0.555 | 0.562 | 0.544 | 0.549 | 0.319 | 0.632 |
| Cafe | PSNR↑ | 24.715 | 19.428 | 28.846 | 24.315 | 29.283 | 24.838 | 22.020 | 19.440 | 22.198 | 20.506 | 25.086 | 14.251 |
| | SSIM↑ | 0.788 | 0.712 | 0.902 | 0.803 | 0.905 | 0.813 | 0.627 | 0.590 | 0.637 | 0.608 | 0.780 | 0.448 |
| | LPIPS↓ | 0.171 | 0.214 | 0.087 | 0.128 | 0.108 | 0.161 | 0.452 | 0.474 | 0.440 | 0.452 | 0.229 | 0.579 |
| Canteen | PSNR↑ | 23.211 | 19.077 | 27.318 | 21.632 | 27.335 | 22.159 | 21.103 | 18.558 | 21.116 | 18.476 | 24.360 | 12.861 |
| | SSIM↑ | 0.733 | 0.631 | 0.849 | 0.712 | 0.838 | 0.734 | 0.591 | 0.546 | 0.592 | 0.540 | 0.761 | 0.426 |
| | LPIPS↓ | 0.253 | 0.330 | 0.168 | 0.236 | 0.204 | 0.263 | 0.483 | 0.507 | 0.480 | 0.505 | 0.225 | 0.595 |
| Center | PSNR↑ | 24.677 | 21.801 | 28.728 | 24.264 | 30.035 | 25.802 | 21.641 | 18.870 | 21.953 | 19.468 | 25.098 | 14.574 |
| | SSIM↑ | 0.754 | 0.696 | 0.848 | 0.763 | 0.872 | 0.813 | 0.598 | 0.559 | 0.609 | 0.564 | 0.737 | 0.486 |
| | LPIPS↓ | 0.239 | 0.282 | 0.170 | 0.213 | 0.169 | 0.203 | 0.489 | 0.524 | 0.475 | 0.507 | 0.288 | 0.607 |
| Innovation | PSNR↑ | 24.258 | 22.062 | 28.980 | 25.201 | 30.554 | 26.390 | 21.964 | 21.357 | 22.021 | 21.525 | 24.518 | 14.389 |
| | SSIM↑ | 0.712 | 0.677 | 0.858 | 0.771 | 0.898 | 0.819 | 0.573 | 0.568 | 0.574 | 0.570 | 0.687 | 0.424 |
| | LPIPS↓ | 0.250 | 0.269 | 0.137 | 0.164 | 0.120 | 0.148 | 0.440 | 0.445 | 0.438 | 0.440 | 0.308 | 0.558 |
| Lab | PSNR↑ | 24.924 | 22.003 | 31.651 | 27.325 | 32.890 | 28.875 | 23.614 | 22.889 | 23.624 | 22.873 | 25.840 | 15.565 |
| | SSIM↑ | 0.824 | 0.785 | 0.926 | 0.869 | 0.939 | 0.898 | 0.725 | 0.715 | 0.725 | 0.716 | 0.812 | 0.544 |
| | LPIPS↓ | 0.145 | 0.167 | 0.069 | 0.093 | 0.066 | 0.087 | 0.351 | 0.361 | 0.360 | 0.371 | 0.198 | 0.468 |
| Library | PSNR↑ | 25.103 | 22.427 | 29.192 | 25.137 | 30.137 | 26.250 | 23.796 | 22.830 | 23.794 | 22.883 | 25.779 | 15.446 |
| | SSIM↑ | 0.671 | 0.620 | 0.782 | 0.699 | 0.806 | 0.746 | 0.589 | 0.574 | 0.589 | 0.574 | 0.692 | 0.417 |
| | LPIPS↓ | 0.286 | 0.324 | 0.209 | 0.249 | 0.206 | 0.243 | 0.423 | 0.435 | 0.423 | 0.435 | 0.260 | 0.585 |

| On 360Roam | | OmniGS | | SC-OmniGS | | SC-OmniGS | | BARF° | | L2G-NeRF° | | CamP° | |
|---|---|---|---|---|---|---|---|---|---|---|---|---|---|
| | | train | test | train | test | train | test | train | test | train | test | train | test |
| | Perturb | ✓ | | ✓ | | ✓ | | ✓ | | ✓ | | ✓ | |
| | Point Init | SfM | | Random | | SfM | | N/A | | N/A | | N/A | |
| Bar | PSNR↑ | 18.915 | 14.718 | 24.876 | 22.090 | 25.410 | 22.556 | 19.457 | 18.499 | 19.803 | 18.794 | 22.946 | 12.600 |
| | SSIM↑ | 0.640 | 0.431 | 0.840 | 0.763 | 0.854 | 0.785 | 0.519 | 0.498 | 0.533 | 0.510 | 0.765 | 0.380 |
| | LPIPS↓ | 0.404 | 0.504 | 0.192 | 0.235 | 0.166 | 0.205 | 0.567 | 0.580 | 0.542 | 0.557 | 0.273 | 0.636 |
| Base | PSNR↑ | 21.449 | 14.559 | 28.322 | 24.780 | 29.226 | 24.308 | 20.986 | 20.024 | 21.382 | 20.122 | 25.179 | 13.251 |
| | SSIM↑ | 0.674 | 0.351 | 0.842 | 0.777 | 0.880 | 0.777 | 0.488 | 0.472 | 0.501 | 0.481 | 0.728 | 0.381 |
| | LPIPS↓ | 0.328 | 0.498 | 0.172 | 0.198 | 0.114 | 0.157 | 0.590 | 0.601 | 0.557 | 0.572 | 0.282 | 0.653 |
| Cafe | PSNR↑ | 22.313 | 15.680 | 28.156 | 23.917 | 29.278 | 25.171 | 22.169 | 19.895 | 22.518 | 20.146 | 26.908 | 13.689 |
| | SSIM↑ | 0.734 | 0.441 | 0.894 | 0.789 | 0.904 | 0.827 | 0.607 | 0.563 | 0.617 | 0.571 | 0.829 | 0.429 |
| | LPIPS↓ | 0.294 | 0.462 | 0.123 | 0.178 | 0.108 | 0.145 | 0.478 | 0.497 | 0.454 | 0.479 | 0.196 | 0.620 |
| Canteen | PSNR↑ | 22.814 | 14.273 | 27.494 | 21.251 | 27.259 | 22.139 | 21.395 | 18.887 | 21.761 | 17.027 | 26.388 | 12.691 |
| | SSIM↑ | 0.732 | 0.458 | 0.844 | 0.692 | 0.837 | 0.732 | 0.564 | 0.521 | 0.575 | 0.476 | 0.817 | 0.445 |
| | LPIPS↓ | 0.331 | 0.536 | 0.198 | 0.289 | 0.206 | 0.265 | 0.526 | 0.558 | 0.503 | 0.571 | 0.196 | 0.627 |
| Center | PSNR↑ | 22.740 | 15.597 | 28.972 | 24.482 | 29.706 | 25.304 | 22.275 | 19.689 | 22.859 | 16.855 | 26.616 | 14.471 |
| | SSIM↑ | 0.717 | 0.510 | 0.847 | 0.779 | 0.867 | 0.799 | 0.584 | 0.524 | 0.604 | 0.478 | 0.780 | 0.487 |
| | LPIPS↓ | 0.372 | 0.553 | 0.205 | 0.265 | 0.177 | 0.220 | 0.505 | 0.540 | 0.474 | 0.559 | 0.264 | 0.608 |
| Innovation | PSNR↑ | 21.880 | 16.047 | 28.916 | 25.943 | 30.079 | 24.788 | 22.291 | 21.242 | 22.761 | 21.450 | 25.890 | 13.361 |
| | SSIM↑ | 0.697 | 0.440 | 0.828 | 0.785 | 0.887 | 0.762 | 0.545 | 0.535 | 0.558 | 0.545 | 0.738 | 0.421 |
| | LPIPS↓ | 0.325 | 0.447 | 0.199 | 0.219 | 0.129 | 0.177 | 0.475 | 0.482 | 0.449 | 0.460 | 0.287 | 0.616 |
| Lab | PSNR↑ | 22.049 | 16.642 | 32.175 | 27.568 | 32.801 | 28.812 | 23.997 | 22.838 | 24.622 | 22.951 | 27.002 | 14.315 |
| | SSIM↑ | 0.762 | 0.563 | 0.930 | 0.874 | 0.938 | 0.895 | 0.709 | 0.692 | 0.729 | 0.707 | 0.837 | 0.530 |
| | LPIPS↓ | 0.299 | 0.421 | 0.089 | 0.122 | 0.068 | 0.090 | 0.361 | 0.372 | 0.314 | 0.330 | 0.209 | 0.594 |
| Library | PSNR↑ | 24.725 | 17.437 | 29.588 | 24.710 | 30.095 | 26.202 | 24.514 | 22.796 | 24.944 | 22.838 | 28.141 | 14.891 |
| | SSIM↑ | 0.684 | 0.445 | 0.791 | 0.703 | 0.805 | 0.743 | 0.584 | 0.558 | 0.600 | 0.568 | 0.798 | 0.422 |
| | LPIPS↓ | 0.323 | 0.495 | 0.225 | 0.289 | 0.207 | 0.244 | 0.430 | 0.451 | 0.402 | 0.430 | 0.205 | 0.623 |

Table 7: The training camera pose quantitative evaluation among calibration methods. Checked "Perturb" indicates perturbed camera poses as inputs, † indicates training from scratch, "Point Init" indicates the way of point cloud initialization for 3D-GS based methods, "$\mathbf{p}$" and "$R$" indicate Root Mean Squared Error (RMSE) of camera position (in world units) and rotation (in degrees), respectively. Methods marked with superscript ° are modified via omnidirectional sampling. SC-OmniGS performs robust camera calibration capability in different scenarios and point initialization.

| On 360Roam | | BARF | BARF° | L2G-NeRF | L2G-NeRF° | CamP | CamP° | SC-OmniGS | SC-OmniGS |
|---|---|---|---|---|---|---|---|---|---|
| Perturb | | ✓ | ✓ | ✓ | ✓ | ✓ | ✓ | ✓ | ✓ |
| Point Init | | N/A | N/A | N/A | N/A | N/A | N/A | random | SfM |
| Bar | $\mathbf{p}\downarrow$ | 0.31873 | 0.11240 | 0.23656 | 0.05947 | 0.16559 | 0.16692 | 0.03811 | 0.03401 |
| | $R\downarrow$ | 0.12260 | 0.03499 | 0.08151 | 0.06093 | 0.02700 | 0.02568 | 0.01880 | 0.01402 |
| Base | $\mathbf{p}\downarrow$ | 0.38139 | 0.03944 | 0.22836 | 0.18336 | 0.19603 | 0.19792 | 0.08044 | 0.02074 |
| | $R\downarrow$ | 0.10911 | 0.00561 | 0.05018 | 0.02207 | 0.02758 | 0.02575 | 0.02459 | 0.00318 |
| Cafe | $\mathbf{p}\downarrow$ | 0.34125 | 0.32115 | 0.14891 | 0.18808 | 0.14154 | 0.14064 | 0.00651 | 0.00627 |
| | $R\downarrow$ | 0.12002 | 0.08143 | 0.03887 | 0.07296 | 0.02560 | 0.02694 | 0.00236 | 0.00212 |
| Canteen | $\mathbf{p}\downarrow$ | 0.47954 | 0.24846 | 0.55104 | 0.58446 | 0.16421 | 0.16661 | 0.04292 | 0.03002 |
| | $R\downarrow$ | 0.18377 | 0.09021 | 0.23060 | 0.18187 | 0.02624 | 0.02444 | 0.00592 | 0.00253 |
| Center | $\mathbf{p}\downarrow$ | 0.72546 | 0.53148 | 0.72888 | 0.81537 | 0.19709 | 0.19951 | 0.17692 | 0.10194 |
| | $R\downarrow$ | 0.26783 | 0.19900 | 0.22620 | 0.38847 | 0.02768 | 0.02532 | 0.06964 | 0.00746 |
| Innovation | $\mathbf{p}\downarrow$ | 0.23938 | 0.20665 | 0.11435 | 0.30508 | 0.20174 | 0.20299 | 0.00565 | 0.02205 |
| | $R\downarrow$ | 0.08755 | 0.06569 | 0.03044 | 0.06525 | 0.02823 | 0.025232 | 0.00190 | 0.00598 |
| Lab | $\mathbf{p}\downarrow$ | 0.07353 | 0.02230 | 0.03886 | 0.01235 | 0.23800 | 0.23774 | 0.01353 | 0.01432 |
| | $R\downarrow$ | 0.02864 | 0.00385 | 0.01433 | 0.00301 | 0.03342 | 0.02524 | 0.00248 | 0.00191 |
| Library | $\mathbf{p}\downarrow$ | 0.27276 | 0.02723 | 0.26827 | 0.02759 | 0.21650 | 0.21446 | 0.11948 | 0.00632 |
| | $R\downarrow$ | 0.07719 | 0.00248 | 0.07728 | 0.00283 | 0.02787 | 0.02771 | 0.01251 | 0.00162 |

| On OmniBlender | | BARF | BARF° | L2G-NeRF | L2G-NeRF° | CamP | CamP° | SC-OmniGS | SC-OmniGS | SC-OmniGS |
|---|---|---|---|---|---|---|---|---|---|---|
| Perturb | | ✓ | ✓ | ✓ | ✓ | ✓ | ✓ | ✓ | ✓ | ✓ |
| Point Init | | N/A | N/A | N/A | N/A | N/A | N/A | Random | Est. depth | Render depth |
| Barbershop | $\mathbf{p}\downarrow$ | 0.14411 | 0.00053 | 0.00560 | 0.00048 | 0.18435 | 0.17873 | 0.11106 | 0.00032 | 0.00025 |
| | $R\downarrow$ | 0.09418 | 0.00040 | 0.00529 | 0.00047 | 0.08132 | 0.07486 | 0.04919 | 0.00034 | 0.00024 |
| Classroom | $\mathbf{p}\downarrow$ | 0.00882 | 0.00059 | 0.36072 | 0.00062 | 0.21609 | 0.21072 | 0.00015 | 0.00023 | 0.00014 |
| | $R\downarrow$ | 0.00995 | 0.00094 | 0.28451 | 0.00095 | 0.18112 | 0.16902 | 0.00028 | 0.00040 | 0.00021 |
| Flat | $\mathbf{p}\downarrow$ | 0.21386 | 0.00053 | 0.40058 | 0.00048 | 0.25824 | 0.25266 | 0.00051 | 0.00108 | 0.00032 |
| | $R\downarrow$ | 0.15046 | 0.00109 | 0.19573 | 0.00113 | 0.07878 | 0.06339 | 0.00077 | 0.00351 | 0.00035 |

| On OmniBlender | | BARF | BARF° | L2G-NeRF | L2G-NeRF° | CamP | CamP° | SC-OmniGS | SC-OmniGS | SC-OmniGS |
|---|---|---|---|---|---|---|---|---|---|---|
| Perturb | | † | † | † | † | † | † | † | † | † |
| Point Init | | N/A | N/A | N/A | N/A | N/A | N/A | Random | Est. depth | Render depth |
| Barbershop | $\mathbf{p}\downarrow$ | 0.34757 | 0.00065 | 0.37682 | 0.00050 | 0.41992 | 0.11743 | 0.00126 | 0.00061 | 0.00037 |
| | $R\downarrow$ | 0.30309 | 0.00058 | 0.24394 | 0.00047 | 0.07589 | 0.25327 | 0.00202 | 0.00065 | 0.00059 |
| Classroom | $\mathbf{p}\downarrow$ | 0.45917 | 0.00041 | 0.41830 | 0.00055 | 0.49153 | 0.33876 | 0.00071 | 0.00064 | 0.00018 |
| | $R\downarrow$ | 0.34051 | 0.00061 | 0.30008 | 0.00096 | 0.25800 | 0.51458 | 0.00093 | 0.00111 | 0.00018 |
| Flat | $\mathbf{p}\downarrow$ | 0.31282 | 0.00050 | 0.39268 | 0.00034 | 0.27143 | 0.00096 | 0.00308 | 0.00093 | 0.00060 |
| | $R\downarrow$ | 0.21171 | 0.00045 | 0.23691 | 0.00044 | 0.15632 | 0.01593 | 0.00883 | 0.00171 | 0.00088 |

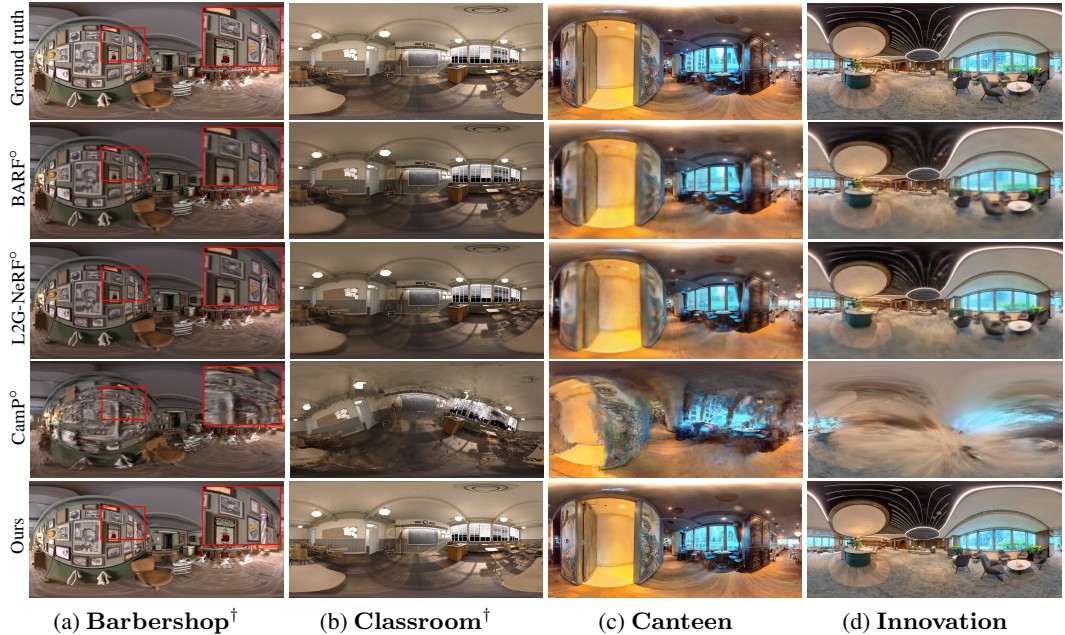

Figure 7: Qualitative comparisons of 360-degree novel views among calibration methods equipped with omnidirectional sampling. Our results outperform in both rendering quality and camera accuracy. † indicates training from scratch, ° indicates baselines modified via omnidirectional sampling.

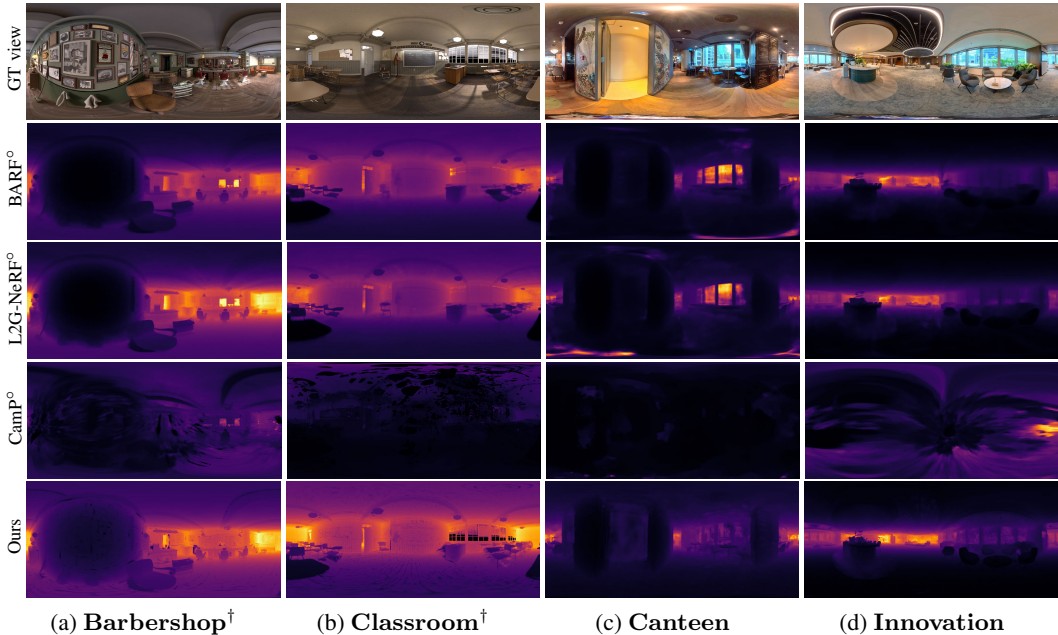

Figure 8: Depth visualization of 360-degree views rendered by calibration methods equipped with omnidirectional sampling. Our results outperform in geometry accuracy and details. † indicates training from scratch, ° indicates baselines modified via omnidirectional sampling.

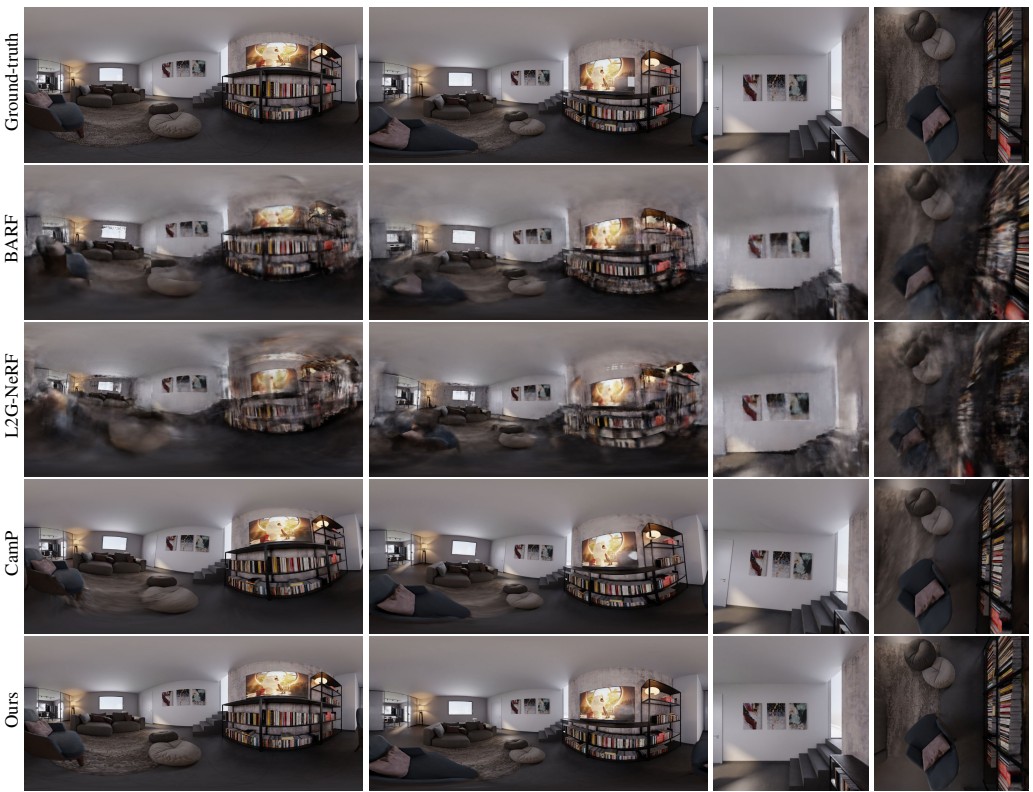

Figure 9: Novel views on synthetic scene **Flat** among baselines trained from scratch.

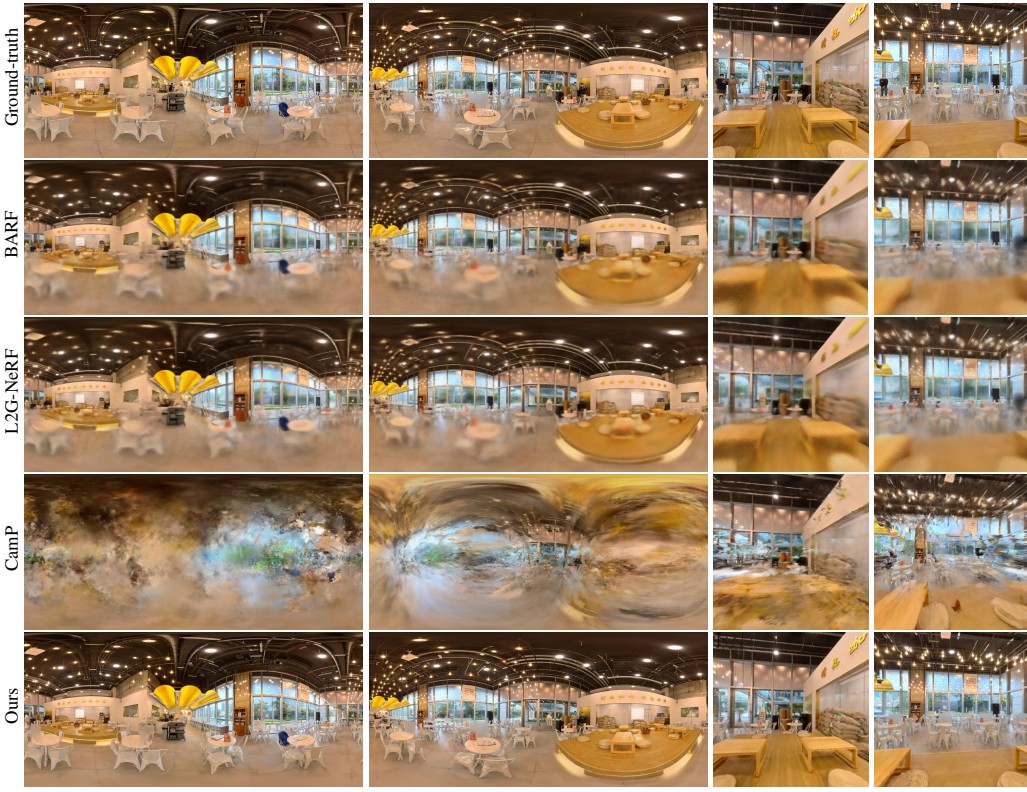

Figure 10: Novel views on real scene **Cafe** among baselines trained with camera perturbation.

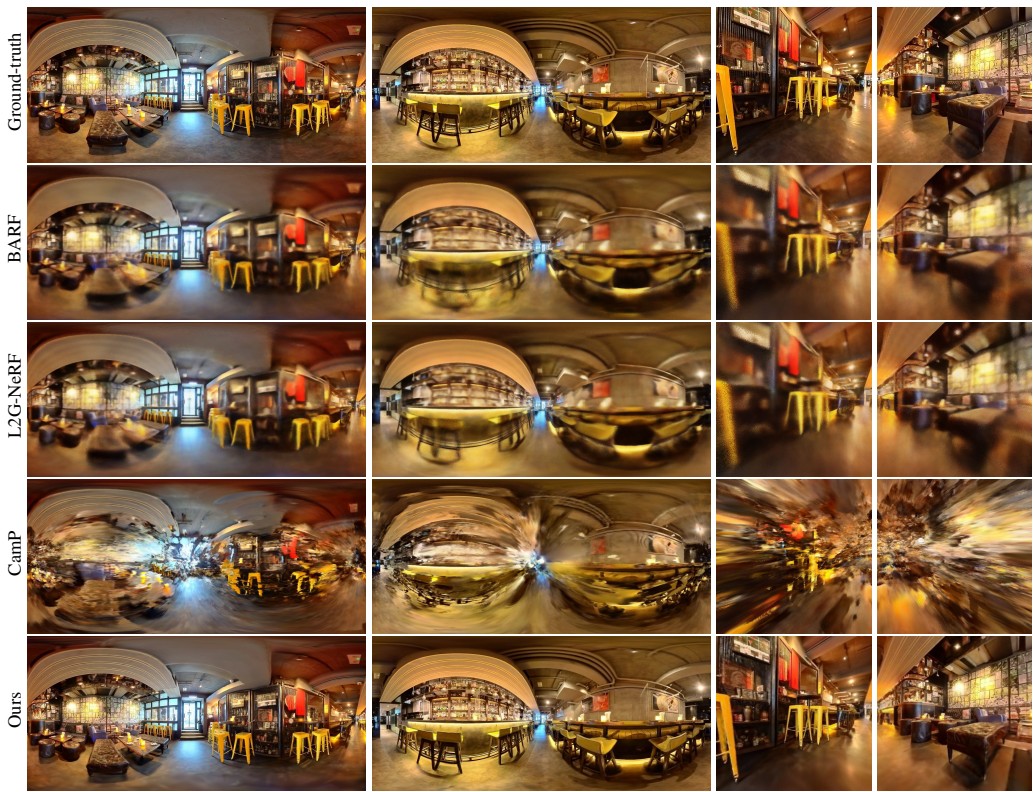

Figure 11: Novel views on real scene **Bar** among baselines trained with camera perturbation.

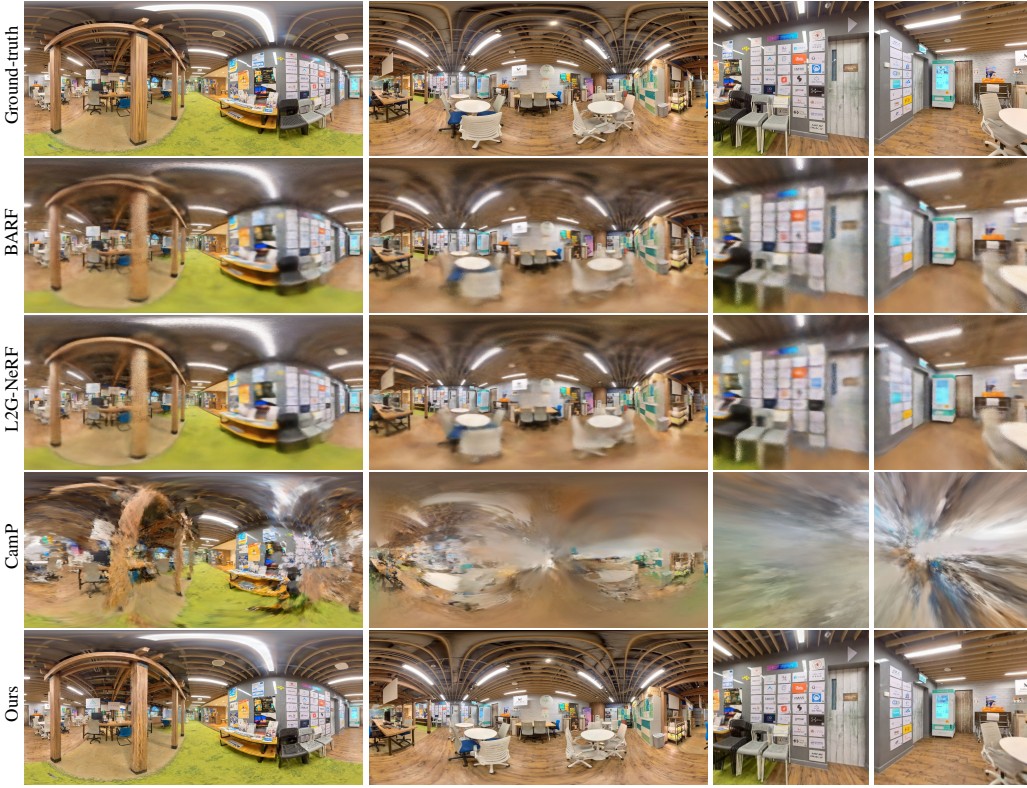

Figure 12: Novel views on real scene **Base** among baselines trained with camera perturbation.

