# OpenReview forum: "SC-OmniGS: Self-Calibrating Omnidirectional Gaussian Splatting"
_ICLR.cc/2025/Conference — ICLR 2025 Poster_

### Official Review · Reviewer_wcnp · 2024-10-23

**Soundness:** 3
**Presentation:** 3
**Contribution:** 3
**Rating:** 8
**Confidence:** 4

**Summary:**

The authors introduce the first system capable of self-calibrating omnidirectional radiance fields, by simultaneously optimizing 3D Gaussians, omnidirectional camera poses, and camera models. Unlike previous works that project  360-degree images onto cube maps, this study preserves the integrity of 360-degree images by directly modeling the 360-degree camera. Additionally, this approach does not rely on precise camera calibration, allowing it to flexibly adapt to various downstream tasks, such as omnidirectional SLAM, by optimizing both intrinsic and extrinsic 360-degree camera parameters. This method achieves the best results among omnidirectional and self-calibration approaches based on NeRF and Gaussians.

**Strengths:**

1. The motivation of the paper is sound. Omnidirectional images have varying information densities across different pixels. Using cube maps-based projection combined with traditional perspective analysis is bound to introduce distortion (or aliasing). By directly modeling the 360-degree camera projection, the method leverages the continuity of 3D space, which can mitigate distortion and provide higher fidelity.
2. I like the chain rule-based derivation of the pose gradient in Equations 13 and 14, as it makes the camera optimization more reasonable.
3. The experiments are very thorough. The authors compare two datasets and cutting-edge methods (OmniGS), while also discussing the experimental results under different camera and point cloud initializations (Tab. 1), as well as the results after adding various perturbations (Fig. 5). This supports it to be a well-rounded work.

**Weaknesses:**

1. Some figure captions are too brief (Fig.2, Fig.3), requiring readers to refer back to the main text for clarification on several unclear points, which disrupts the reading flow.
2. In the visualizations, it seems that only the rendered results are compared, lacking some geometric information, such as depth visualizations and point cloud reconstructions.

**Questions:**

I have some confusion regarding the input. I notice that in datasets like the 360Roam dataset, there are 110 training views and 37 test views. Are these views entirely omnidirectional data, or are there some additional perspective camera images used as auxiliary data?

---

> ### Author Response · Authors · 2024-11-24
>
> >***W1: Brief captions in Figure 2 and 3.***
>
> We tried to elaborate figure details in the captions while the space exceeded the limitation. Considering these two figures are self-explanatory and their corresponding contents are on the same page, we therefore keep captions of Figure 2 and 3 concise.
>
> >***W2: Lack some geometric information, such as depth visualizations and point cloud reconstructions.***
>
> Thank you for your suggestions. We have added depth visualizations to the Appendix of the revised version. Please refer to Figure 8 of the revision.
>
> >***Q1: About the benchmark dataset.***
>
> Yes, the dataset 360Roam and Omniblender are only composed of 360-degree images and commonly used to evaluate omnidirectional radiance field methods performance.

---

> > ### Author Response · Authors · 2024-11-26
> >
> > Dear Reviewer,
> >
> > Thank you very much for dedicating your time and effort to reviewing our paper. We are grateful for your constructive feedback, which has significantly enhanced the quality of our work.
> >
> > If you have any further concerns or suggestions, please do not hesitate to share them with us. We look forward to the opportunity for further discussion and paper refinement. And we hope that our work can make a valuable contribution to the community.
> >
> > Best regards,
> >
> > The Authors

---

> > > ### Comment · Reviewer_wcnp · 2024-11-26
> > > **Thank you for the answering**
> > >
> > > I have looked through all the rebuttal comments, including those of other reviewers. They're arranged in detail and have fully convinced me. There's no more concerns and I decide to raise my score.

---

### Official Review · Reviewer_dnTE · 2024-10-31

**Soundness:** 3
**Presentation:** 4
**Contribution:** 3
**Rating:** 6
**Confidence:** 4

**Summary:**

The paper proposes a self-calibrating Gaussian splatting method for reconstructing omnidirectional radiance fields from 360-degree images without poses or with noisy poses. In this framework, scene representation, camera poses, and camera models are jointly optimized by minimizing a weighted spherical photometric loss. Additionally, a differentiable omnidirectional camera model is introduced to learn camera distortion. Experimental results show that the proposed method effectively recovers high-quality radiance fields from 360-degree image inputs.

**Strengths:**

(1) The paper proposes a novel self-calibrating method that extends the omnidirectional Gaussian splatting to handle unposed or noisy 360-degree images.

(2) The paper introduces a differentiable omnidirectional camera model, which uses trainable focal length and angle distortion coefficients to represent camera distortion.

(3) The proposed method achieves state-of-the-art performance in novel view synthesis.

**Weaknesses:**

(1) The method estimates camera poses but lacks comparisons of pose accuracy. I think that comparing only rendering quality, especially with NeRF-based calibration methods, is insufficient to determine whether the superior performance of this paper is due to pose optimization, the camera model, or the scene representation using 3D GS. Therefore, I suggest adding quantitative and qualitative comparisons of camera poses on two datasets.

(2) The experiment only compared with NeRF-based calibration methods and lacks comparisons with 3D GS-based baselines, such as COLMAP-free 3D GS. Besides, these NeRF-based calibration methods are originally designed to address noisy poses, rather than unposed images. I think it would be fairer to compare with poes-prior-free methods, such as NoPE-NeRF or LocalRF.

References:

A1. Fu, Y., Liu, S., Kulkarni, A., et al. COLMAP-Free 3D Gaussian Splatting, CVPR, 2024.

A2. Bian, W., Wang, Z., Li, K. et al. Nope-NeRF: Optimising neural radiance field with no pose prior, CVPR, 2023.

A3. Meuleman A, Liu Y L, Gao C, et al. Progressively optimized local radiance fields for robust view synthesis, CVPR, 2023.

(3) The paper does not include an ablation study of the anisotropy regularizer loss.

**Questions:**

(1) It would be better to provide the experimental results mentioned in the weaknesses.

(2) In Table 1, the paper does not provide results with random initialization and estimated depth in the comparison of perturbed poses.

(3) The paper states that it can address scenes with wide baselines. However, the method cannot be trained from scratch on real-world multi-room scenes, even though real-world datasets have more views than synthetic datasets. I recommend including a detailed analysis and results of the failure cases to better understand the reasons behind this limitation.

---

> ### Author Response · Authors · 2024-11-24
>
> >***W1: Lack comparisons of camera poses.***
>
> Thank you for your suggestion. We have added the evaluation results of the camera pose into the revised version, including Table 7. Our method indeed achieves highest pose estimation accuracy leading to high-fidelity radiance field reconstruction.
>
> >***W2: Lack comparison with poes-prior-free methods, e.g. COLMAP-free 3D GS, NoPE-NeRF.***
>
> _Comparisons with calibration methods with no pose prior. Note that either Nope-NeRF or CF-3DGS requires depth prior during camera calibration, but our SC-OmniGS calibrates camera without any depth prior during optimization.* was reported in the paper._
>
> | OmniBlender (test) | Perturb |  Point Init | Barbershop | Classroom | Flat |
> |---|---|---|---|---|---|
> |  |  |  | PSNR / SSIM / LPIPS |  PSNR / SSIM / LPIPS | PSNR / SSIM / LPIPS |
> | Nope-NeRF | † | N/A | 14.113 /  0.451 /  0.685  | 16.911 / 0.619 / 0.712 |   12.760  /  0.586 /  0.652  |
> | CF-3DGS | † | est. depth |  15.635 / 0.501 / 0.510 | 14.823 / 0.539 / 0.612 |14.586 / 0.597 / 0.479 |
> | SC-OmniGS* | † | random | 33.422 / 0.944 / 0.084 | 28.971 / 0.806 / 0.214 | 31.673 / 0.895 / 0.114 |
> | SC-OmniGS* | † | est. depth|  33.401 / 0.940 / 0.087 | 29.385 / 0.801 / 0.195 | 31.278 / 0.897 / 0.094 |
> ||||||
>
> Although some self-calibrating methods emphasize they are pose-prior-free in their papers, they in fact only support object-centric scenes or using sequential data as input.
>
> NoPE-NeRF and COLMAP-free 3DGS heavily rely on depth supervision during the training process which make them sensitive to depth prior obtained from monocular depth estimation models. By contrast, our method only uses coarse point clouds for 3D Gaussians initialization without relying on dense depth supervision. Even though using random point cloud initialization, our methods still have dominant performance, demonstrating our robustness and flexibility.
>
> Moreover,  A1 COLMAP-free 3DGS and A3 [Meuleman A et al 2023] should be closely relevant to radiance-field-based SLAM methods which require videos as input and make use of sequential relationships to progressively recover radiance fields. The incoming frames can be roughly initialized by the motion model. However, the benchmark dataset for omnidirectional radiance field evaluation is composed of sparse and discrete frames. Therefore, we did not use them as baselines in the paper.
>
> >***W3: Lack ablation study of the anisotropy regularizer loss.***
>
> Since anisotropy regularizer has become a practical use and is not counted as our contribution, we did not conduct an ablation study to further verify its effectiveness in our paper.
>
> >***Q1: It would be better to provide the experimental results mentioned in the weaknesses.***
>
> Please refer to the responses in W1-3.
>
> >***Q2: Table 1 lacks results of the proposed methods when the input camera is perturbed while 3D gaussians are initialized randomly or from estimated depth.***
>
> Thank you for your thoughtful suggestion. We have reported these results in Table 1 of the revised version.
>
>
> >***Q3: Detailed analysis and results of the failure cases on real-world multi-room scenes to better understand the reasons behind this limitation.***
>
> As we discussed in the “Limitations” of the paper (Line535-539), all self-calibration methods fail to learn radiance fields without any pose priors in challenging multi-room-level scenes. This is because the tolerance level of the self-calibrating radiance field method is capped. We have analyzed our method robustness against varying levels of camera perturbation in Sec. 5.4. With the noise increasing to some levels, the reconstruction performance drops obviously. Still, our SC-OmniGS consistently outperforms baselines.
>
> When training from scratch without pose prior, the initial camera poses are identical, at the origin of world coordinates. Therefore, in challenging cases, i.e,  sparse and discrete views on multi-room-scale scenes, there is no doubt that the noise of initial value has exceeded tolerance level.

---

> > ### Comment · Reviewer_dnTE · 2024-11-26
> > **Thanks for the rebuttal comments**
> >
> > Thank you for your responses to the comments. I am pleased to note that the revised manuscript has addressed my primary concerns regarding the evaluation of pose errors and the comparison with additional methods. The proposed method demonstrates good performance in camera calibration and scene reconstruction. As a result, I lean toward accepting the paper and giving my final rating as 6.

---

> ### Author Response · Authors · 2024-11-26
>
> Dear Reviewer,
>
> Thank you very much for dedicating your time and effort to reviewing our paper. We are grateful for your constructive feedback, which has significantly enhanced the quality of our work.
>
> If you have any further concerns or suggestions, please do not hesitate to share them with us. We look forward to the opportunity for further discussion and paper refinement. And we hope that our work can make a valuable contribution to the community.
>
> Best regards,
>
> The Authors

---

### Official Review · Reviewer_83HV · 2024-10-31

**Soundness:** 3
**Presentation:** 4
**Contribution:** 3
**Rating:** 8
**Confidence:** 3

**Summary:**

This paper proposes a system for self-calibrating omnidirectional radiance fields, aiming to optimize 3D Gaussians, omnidirectional camera poses, and camera models in tandem. While the authors describe this as the first system of its kind, the contribution can largely be seen as an engineering effort to integrate multiple optimized parameters within a single framework. The primary novelty in the work appears to lie in the introduction of a differentiable omnidirectional camera model that enables ray-wise distortion handling and in the derivation of gradients for pose optimization.

**Strengths:**

- The paper is well-structured and easy to follow, making the methodology and findings accessible to readers.
- The introduction of a differentiable omnidirectional camera model that enables ray-wise distortion and the derivation of gradients for pose optimization are valuable innovations that expand the applicability of the system.
- The use of spherical weights in the photometric loss ensures spatially balanced optimization, which enhances the robustness and accuracy of the optimization process.
- The experiments and results demonstrate superior performance compared to previous methods, highlighting the effectiveness of the proposed approach.

**Weaknesses:**

Misleading Terminology in Title: While the title suggests "self-calibrating omnidirectional Gaussian splatting," the approach relies on initialization from a structure-from-motion (SfM) pipeline rather than directly calibrating intrinsic parameters from the images alone. This approach is more accurately an optimization process rather than an auto-calibration technique in the classical sense (e.g., auto-calibration from absolute dual quadrics in multiple-view geometry).

**Questions:**

- The paper reports PSNR results to demonstrate performance, but since it also optimizes camera poses, it would be beneficial to include a comparison of the optimized extrinsic parameters against ground truth values.

- Given the emphasis on self-calibration, could the paper also show improvements in the intrinsic parameters after optimization?

- Are the optimizations of focal length and distortion parameters shared across all views, or are they optimized per view? Clarifying this would be help.

---

> ### Author Response · Authors · 2024-11-24
>
> Thank you for the precise and insightful comments, please find our responses below:
> >***W1: About title preciseness of self-calibrating.***
>
> This point is worth discussion. As defined, calibration is to determine or adjust the accuracy and quality of measurements. Given SfM estimation without perturbation, SC-OmniGS can continue to refine camera models and poses which increases reconstruction performance, as evidenced in Table 2. In our experiments, we have also studied various situations of radiance field self-calibration involving varying levels of initialization noise to verify our method's robustness. Training from scratch is just a special situation where all camera poses are initialized at the origin of world coordinates. In essence, our paper title, "SC-OmniGS: Self-Calibrating Omnidirectional Gaussian Splatting," accurately encapsulates the core content of our research.
>
> >***Q1 Evaluation of optimized camera poses.***
>
> We appreciate your suggestion and have incorporated the evaluation results of optimized camera poses in the revised version. Our method has indeed achieved the highest accuracy in pose estimation, a fundamental aspect crucial for the reconstruction of high-fidelity radiance fields. For details, please refer to Table 7 in the revised paper.
>
> >***Q2: Showcase of improvements in the intrinsic parameters.***
>
> In some self-calibrating radiance field papers, they utilize Colmap results as ground truth to evaluate their optimized intrinsic parameters. However, as we mentioned in the introduction section (i.e., Line 43-46), existing SfM methods rely on an idealized omnidirectional camera model assumption and overlook the adverse effects of omnidirectional camera distortion in real-world scenarios. We are the first to tackle this issue but cannot obtain pseudo ground truth to conduct quantitative evaluation of intrinsic parameters. As a radiance field method, we instead relied on reconstruction quality, i.e. novel view rendering quality to reflect the improvement of camera parameters. We also conducted an ablation study (Table 3) to solely evaluate camera model efficacy in terms of reconstruction quality. We believe the experiment is comprehensive.
>
> >***Q3: Are the optimizations of focal length and distortion parameters shared across all views?***
>
> Yes, we only optimized a single omnidirectional camera model and used it to tackle all views on each scene. We have made it clearer in the implementation details of the revised version (Line 335-336).

---

> > ### Comment · Reviewer_83HV · 2024-11-26
> > **Thanks for the rebuttal**
> >
> > Thanks for clarification, I will keep the score same and haven't got any further concern.

---

### Official Review · Reviewer_CA85 · 2024-11-02

**Soundness:** 3
**Presentation:** 2
**Contribution:** 3
**Rating:** 6
**Confidence:** 3

**Summary:**

This paper proposes an extension of 3D Gaussian splatting (GS) for omnidirectional images and enabling self-calibration. GS for omnidirectional images is previously studied (OmniGS), but it assumes the pre-computed camera positions. The proposed method refines the camera poses during gradient-based optimization. Experiments show that the proposed method improves the vanilla OmniGS. A main technical contribution of the paper is to derive the backward gradient for pose refinement of spherical images.

**Strengths:**

### Self calibration + omnidirectional images
The proposed method would be the first attempt to combine the self-calibration and omnidirectional GS.

### Backward gradient for spherical images
For pose refinement of spherical images, the paper derives the gradients in Eqs. (13--14). This would be a technical novelty in this paper.

**Weaknesses:**

### Limited technical improvement
Self-calibration of GS has been well-studied so far. Also, omnidirectional GS is existing. The proposed system would be practical, but the scientific motivation for combining those two is not quite large, i.e., the technical novelty is limited.

### Gradient derivation
The key technical part of the paper, the derivation of gradients on camera poses of spherical images (Eqs. (13--14)), is rather straightforward. This would be naturally extended from perspective cases.

While the paper describes "converting 360-degree images to cube maps..." this is just about the camera models.
Although in different contexts, for example, Metashape and OpenMVS support spherical camera models for the SfM problem, which involves bundle adjustment (i.e., non-linear optimization using first-order derivative), so they should compute gradients in somewhat similar ways to Eqs. (13--14).

**Questions:**

I would appreciate it if the authors emphasized the technical (or scientific) novelty of the proposed method again.

---

> ### Author Response · Authors · 2024-11-24
>
> >***W1: Limited technical improvement.***
> The scientific motivation for combining Self-calibration and omnidirectional GS is limited.
>
> The proposed SC-OmniGS is not an incremental work that simply combines existing solutions. Although the pose optimization along Gaussian splatting process has been studied in some recent GS-based SLAM methods, they only support perspective images without distortion. Their theoretical analysis and implementation of camera pose derivatives in 3D Gaussian Splatting cannot be directly reused to achieve self-calibrating omnidirectional radiance fields. Additionally, joint optimization of intrinsic camera models and GS is still underexplored.
>
> Our SC-OmniGS systematically analyzes and achieves omnidirectional camera pose optimization within the omnidirectional Gaussian splatting procedure. We are the first to tackle the complex distortion pattern contained in omnidirectional cameras via introducing a novel differentiable omnidirectional camera model. Furthermore, we proposed a weighted spherical photometric loss to enhance omnidirectional radiance field reconstruction quality.
>
> We believe our work will attract good attention and make a positive contribution to the omnidirectional vision community.
>
> >***W2: Gradient derivation is one of key technical parts of the paper but it has been addressed.***
> The derivation of gradients on spherical camera poses (Eqs. (13--14)) is the key technical part. However it is rather straightforward since it would be naturally extended from perspective cases. Moreover, some SfM softwares supporting the spherical camera already has gradient computing similar to Eqs. (13--14).
>
> The gradient computings in SC-OmniGS and the mentioned softwares (Metashape, OpenMVS) are theoretically different. The mentioned methods optimize camera pose by minimizing 2D-to-3D reprojection residual of corresponding points. The optimization problem is formulated as a factor graph and solved by the Levenberg–Marquardt (LM) algorithm, i.e., the first-order approximation of the error function. By contrast, our optimization objective in SC-OmniGS is to minimize weighted spherical photometric loss between rendering and reference images. The rendering process is differentiable, a key departure from traditional methodologies. The gradients of omnidirectional camera pose are then derived and back-propagated along the GS process.
>
> >***Q1:  Emphasizing technical novelty of the proposed method again.***
>
> The primary technical contributions of our work include:
> 1. **Gradient Derivation for Omnidirectional Camera Poses.** SC-OmniGS stands as a pioneering effort dedicated to the precise calibration of omnidirectional radiance fields, showcasing cutting-edge performance levels. These advancements can further facilitate applications such as GS-based omnidirectional SLAM.
> 2. **Addressing Complex Distortion Patterns with a Generic Omnidirectional Camera Model.** Due to complex distortion patterns inherent in omnidirectional cameras, current 3D omnidirectional vision methods rely on ideal spherical camera model assumption, resulting in suboptimal performance, as we discussed in the main paper Line 43-50. To the best of our knowledge, we are the first to effectively handle this issue by proposing a generic camera model tailored for the 360-degree camera.
> 3. **Enhanced Reconstruction Quality through Weighted Spherical Photometric Loss**. To promote spatially consistent optimization and elevate the overall quality of omnidirectional radiance field reconstruction, we introduce a novel weighted spherical photometric loss function.

---

> > ### Comment · Reviewer_CA85 · 2024-11-25
> > **Thanks for the effective rebuttals**
> >
> > Thanks for the rebuttal comments.
> >
> > I thoroughly went through the others' comments and rebuttals, as well as the revised paper.
> >
> > Regarding the scientific contributions, I may have underestimated them.
> > As the authors mention, this study does have application-oriented contributions, which archive the self-calibration of GS for omnidirectional cameras.
> >
> > I understand the SfM's objectives are reprojection errors (between 2D positions of points), and their gradients are used for LM algorithms.
> > Indeed, this concept is not the same as the GS-like methods, which minimize the photometric error through a differentiable pipeline.
> >
> > I would like to change the rating to a reasonable one.

---

> > > ### Author Response · Authors · 2024-11-25
> > >
> > > Thank you for your thoughtful reconsideration and for increasing the score based on our rebuttal.
> > >
> > > We appreciate the time you took to review our revised paper and the other comments, as well as for recognizing the scientific contributions of our study.

---

### Author Response · Authors · 2024-11-24
**Global Response:**

In the paper, we proposed the first system for self-calibrating omnidirectional radiance fields, which is able to jointly optimize 3D Gaussians, omnidirectional camera poses and camera models. Notably, our work includes a thorough theoretical analysis of omnidirectional camera pose gradients along the omnidirectional Gaussian splatting procedure, allowing efficient and effective optimization of noisy camera poses. Moreover, we introduced a novel differentiable omnidirectional camera model to address the intricate distortion patterns inherent in omnidirectional cameras, thereby enhancing performance in real-world scenarios. The extensive experiments verified that our method achieved state-of-the-art performance.

**We extend our sincere gratitude to all reviewers for their valuable feedback and acknowledgment of excellent presentation, extensive experiments, and the significant contributions our work makes to the research community.**

Please refer to our detailed responses to specific comments provided below. Furthermore, we have carefully revised the manuscript according to reviewers' suggestions, highlighting these changes in magenta. The major modifications in the revised version are summarized below:

1) In Table 1, we have included the evaluation results of SC-OmniGS with point cloud initialization involving both random and estimated depth, given perturbed camera input, in response to Reviewer **dnTE**.
2) In Appendix C.2, we have included the pose optimization evaluation results comparing different calibration methods in Table 7, in response to Reviewers **83HV** and **dnTE**.
3) In Appendix C.2, we have incorporated depth visualizations rendered by various calibration methods in Figure 8, in response to Reviewer **wcnp**.

---

### Meta-Review · Area_Chair_PKNq · 2024-12-20

**Metareview:**

This paper presents a method of self-calibrating 3D Gaussian Splatting from omni-directional images. While there has been a 3DGS method using omni-directional images as input, the paper introduces a differentiable omnidirectional camera model that enables ray-wise distortion handling and develops the derivation of gradients for pose optimization. The strength of the work is on the differentiable omnidirectional camera model, that allows the camera pose/parameter refinement together with the 3DGS updates. The weakness was the lack of evaluation over the optimized camera parameters, basically failing to showcase the strength of the self-calibration part. This part was amended during the interaction between reviewers and authors. As a result, the four expert reviewers were all positive about the paper. The AE agreed with the reviewers' opinions and rendered this recommendation.

**Additional Comments On Reviewer Discussion:**

During the reviewer-author discussion phase, there were questions about the evaluation of the refined camera parameters. The authors clarified the point by an additional table to demonstrate the effectiveness. The reviewers also pointed out that the method was rather incremental without a strong technical novelty. During the discussion, it was agreed that the method actually contained a non-trivial technical contribution in the derivation of the gradient in the omnidirectional camera model, and the method has a strong merit in its application aspect.

---

### Decision · Program_Chairs · 2025-01-22

Accept (Poster)